# CAM: A Constructivist View of Agentic Memory for LLM-Based Reading Comprehension

**Rui Li**[1,*,†,‡]    **Zeyu Zhang**[1,†,‡]    **Xiaohe Bo**[1,†,‡]    **Zihang Tian**[1,†,‡]
**Xu Chen**[1,†,‡,§]    **Quanyu Dai**[2,§]    **Zhenhua Dong**[2]    **Ruiming Tang**[2]
[1]Gaoling School of Artificial Intelligence, Renmin University of China
[2]Huawei Noah's Ark Lab
{lirui121200, xu.chen}@ruc.edu.cn  daiquanyu@huawei.com

## Abstract

Current Large Language Models (LLMs) are confronted with overwhelming information volume when comprehending long-form documents. This challenge raises the imperative of a cohesive *memory* module, which can elevate vanilla LLMs into autonomous reading agents. Despite the emergence of some heuristic approaches, a systematic design principle remains absent. To fill this void, we draw inspiration from *Jean Piaget's Constructivist Theory*, illuminating three traits of the agentic memory—*structured schemata*, *flexible assimilation*, and *dynamic accommodation*. This blueprint forges a clear path toward a more robust and efficient memory system for LLM-based reading comprehension. To this end, we develop **CAM**, a prototype implementation of **Constructivist Agentic Memory** that simultaneously embodies the structurality, flexibility, and dynamicity. At its core, CAM is endowed with an incremental overlapping clustering algorithm for structured memory development, supporting both coherent hierarchical summarization and online batch integration. During inference, CAM adaptively explores the memory structure to activate query-relevant information for contextual response, akin to the human associative process. Compared to existing approaches, our design demonstrates dual advantages in both performance and efficiency across diverse long-text reading comprehension tasks, including question answering, query-based summarization, and claim verification.

## 1 Introduction

Transformer-based Large Language Models (LLMs) [1–4] have emerged as transformative tools in the realm of natural language understanding. Nevertheless, their prowess tends to falter when confronted with extremely long documents [5, 6]. Such a challenge arises not only due to the bounded context capacity of LLMs, but more fundamentally because of the difficulty in perceiving and aggregating critical information fragments dispersed across distant parts of lengthy texts [7, 8].

A prevalent approach to addressing these limitations involves equipping LLMs with *explicit memory modules* [9] for information storage and retrieval, akin to human cognitive patterns. Such modules maintain a cohesive "mental" representation of ingested content, enabling LLMs to function as autonomous

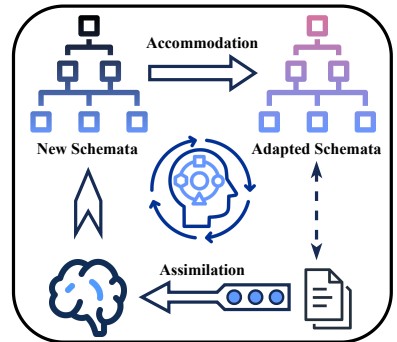

Figure 1: Memory development process under constructivist perspective.

reading agents that can store, retrieve, and reason over long-range dependencies. However, despite the recent proliferation of agentic memory approaches [10–18], they generally mimic human memory at a superficial level, lacking a coherent principle as the basis for their designs. This leads us to raise a critical question: *what are the traits of an effective memory module for LLM-based reading agents?*

---

*Work done during Rui Li's internship at Huawei Noah's Ark Lab. †Beijing Key Laboratory of Research on Large Models and Intelligent Governance. ‡Engineering Research Center of Next-Generation Intelligent Search and Recommendation, MOE. §Corresponding authors.

39th Conference on Neural Information Processing Systems (NeurIPS 2025).

Table 1: Method comparison from the constructivist view. See Appendix B for more details.

| Methods | Type | Structurality | Flexibility | Dynamicity |
|---|---|---|---|---|
| MemGPT [11] | Online | ✗ | ✗ | ✗ |
| ReadAgent [13] | Online | ✗ | ✓ | ✗ |
| RAPTOR [14] | Offline | ✓ | ✓ | ✗ |
| GraphRAG [15] | Offline | ✓ | ✗ | ✗ |
| MemTree [18] | Online | ✓ | ✗ | ✗ |
| **CAM (Ours)** | **Online (Batch)** | ✓ | ✓ | ✓ |

To answer this basic question, we move beyond superficial designs and instead delve into the underpinnings of human cognition. In particular, we draw upon insights from *Jean Piaget's Constructivist Theory* [19–22], a cornerstone of cognitive science that has profoundly shaped the understanding of how human memory develops. As stated in this theory, memory refers to an evolving mental system, which actively organizes the received information into coherent cognitive structures (i.e., ***schemata***). When new information arrives, two fundamental operations (as shown in Figure 1) drive the schemata development process: (1) ***assimilation***, to fit new information into the current memory schemata [23]; and (2) ***accommodation***, to alter the current schemata if new information cannot be readily fitted [22]. These two operations collectively preserve the cognitive equilibration [19–21] of memory schemata, a balanced growth state that facilitates more accurate mental representations of received information. Notably, assimilation exhibits *flexibility*, allowing each information unit to be integrated into multiple relevant locations within the schemata; accommodation exhibits *dynamicity*, enabling the schemata to evolve through localized adjustments without the need for wholesale reconstruction upon every input. Inspired by such a constructivist perspective, we posit three key memory traits to guide the design of LLM-based reading agents—*structured schemata*, *flexible assimilation*, and *dynamic accommodation*.

Current agentic memory systems, however, inherently deviate from such a design principle. As shown in Table 1, early works (e.g., MemGPT [11], MemoryBank [12], and ReadAgent [13]) treat memory as a tabular repository to independently store textual chunks or compressed gists. Such unstructured representations fail to capture the underlying information associations behind long texts. Some recent approaches (e.g. RAPTOR [14], GraphRAG [15], HippoRAG [16], and MemTree [18]) recognize the importance of memory structurality, yet none simultaneously embodies the flexibility and dynamicity during memory structure development. Therefore, we further pursue a new memory implementation that adheres to the constructivist design principle for LLM-based reading agents.

To this end, we present **CAM**, a prototype of **Constructivist Agentic Memory**, aimed at enhancing the long-text reading comprehension capabilities of LLMs. For memory structurality, CAM actively organizes the input documents into a unified hierarchical architecture. At the foundation layer, CAM constructs a coherent semantic network formed by raw text chunks, capturing both textual similarity and narrative coherence. Higher-level memory nodes are abstract summaries aggregated from closely related lower-level nodes. In the process of building this hierarchy, CAM is endowed with *a local-first incremental overlapping clustering algorithm* to support both flexible assimilation (i.e., a lower-level memory node can contribute to multiple higher-level abstractions) and dynamic accommodation (i.e., the memory structure efficiently adapts to new inputs). Notably, our design naturally allows CAM to integrate newly arrived chunks *in batches*, offering significant efficiency advantages (over $4\times$ faster, see Section 5.3) over existing methods that remain confined to offline or entry-wise online settings. At inference time, CAM adopts a *Prune-and-Grow* associative strategy to activate the query-relevant memory nodes for contextual response, delivering high accuracy over diverse long-text reading tasks.

The contributions of this paper are summarized as follows:

- **Blueprint:** Drawing upon Piaget's Constructivist Theory, we present an explicit design principle for agentic memory in LLM-based reading comprehension, highlighting three critical traits of the memory module: structured schemata, flexible assimilation, and dynamic accommodation.

- **Prototype:** We then develop a prototype of Constructivist Agentic Memory (CAM) to enhance the reading comprehension capabilities of LLMs, using an incremental overlapping clustering algorithm for memory development and a Prune-and-Grow strategy for memory retrieval.

- **Evaluation:** We evaluate our design on question answering, query-based summarization, and claim verification benchmarks, covering both single- and multi-document scenarios. The results demonstrate the dual superiority of CAM in performance and efficiency over existing methods.

## 2 Background

**Long-Context LLMs**    Due to the limited context capacity, LLMs struggle with handling long texts. The most direct solution is to fine-tune LLMs with extended context windows [24–29]. Nevertheless, LLM performance tends to decline as input texts lengthen, even when they do not exceed the specified context length [5, 6, 8]. For this issue, many works resort to memory-augmented LLM reading agents.

**Tabular Memory**    Early LLM-based reading agents split lengthy texts into short chunks for memory storage [10–13]. Upon receiving a user query, relevant chunks are recalled to facilitate LLM inference, aligning with the standard Retrieval-Augmented Generation techniques [30–32]. MemoryBank [12] and Ret-LLM [33] manage full historical records with dense retrieval models and read-write functions. MemGPT [11] employs cache-like architectures to prioritize recent information. SCM [34] improves the ability of LLMs to maintain long-term memory through memory stream and controller modules. ReadAgent [13] compresses each text chunk into gist memory, with an interactive look-up process to access related information as needed for specific tasks. While simple to implement, these unstructured memory designs fall short when critical information is dispersed across multiple entries [8, 17].

**Structured Memory**    Recent advances recognize the limitations of unstructured information storage, shifting toward the design of structured memory systems. MemWalker [35] processes the long context into a tree of summary nodes from the bottom up and navigates this tree in search of relevant content for inference. Similarly, RAPTOR [14] also organizes texts into a recursive tree, clustering summaries at each layer to produce higher-level understandings. GraphRAG [15] extracts entities and relations from the context to build a knowledge graph, and generates summaries for groups of related entities. When a question is posed, each summary node provides partial information, which is then combined to form the final answer. GraphReader [36] also transforms the input texts into a graph structure, with a set of predefined functions to facilitate planning and reflection for inference. Although these works effectively structure large-scale textual data to enhance retrieval and generation capabilities, they are confined to static corpora, requiring complete reconstruction to integrate new information. To resolve this, MemTree [18] maintains a dynamic tree-structured memory with an online top-down clustering algorithm. However, it can only integrate new chunks sequentially, with a potential risk of structural imbalance [37]. Moreover, as summarized in Table 1, none of these works simultaneously embodies the flexibility and dynamicity during memory development. See Appendix B for more discussion.

## 3 Blueprint: *Structurality*, *Flexibility*, and *Dynamicity*

As the first step, we seek for an explicit memory design blueprint for LLM-based reading agents. Unlike heuristic approaches in existing literature, we ground our design in a foundational cognitive development framework—Jean Piaget's Constructivist Theory (refer to Appendix A for an overview). Inspired by this theory, we posit three critical memory traits as design objectives: *structured schemata*, *flexible assimilation*, and *dynamic accommodation*. These traits are not mutually exclusive, but rather work in concert to facilitate the integration, retention, and retrieval of information from long texts.

### 3.1 Structured Schemata

In line with Piaget's Constructivist Theory, the memory system should actively restructure all received information into hierarchical schemata from the bottom up. Formally, given a set of input information units (e.g., raw text chunks) $V = \{v_1, v_2, \ldots, v_n\}$, memory first perceives the underlying associations [38] among these basic units to build a foundational semantic network $G_0 = (V, E)$, where $E$ is the set of edges capturing the semantic coherence of unit nodes. Closely related units are then aggregated into higher-level supernodes, forming a coarse network that represents the abstractive understandings. Further aggregation of the abstractions recursively constructs such a hierarchy of memory schemata:

$$M = (\{G_l\}_{l=0}^L, \{\psi_l\}_{l=1}^L), \tag{1}$$

where $G_l = (V_l, E_l)$ represents the graph at level $l$, with $V_l$ and $E_l$ being the set of nodes and edges, respectively; $\psi_l : G_{l-1} \to G_l$ is the upward mapping that reflects the affiliation of low-level elements in $G_{l-1}$ to high-level abstractions in $G_l$. This hierarchical structurality is crucial as it naturally enables the seamless integration of abstract concepts and granular details, fostering a cohesive framework for deep comprehension and accurate recall of complex information. The construction of $M$ are driven by two primary cognitive processes: *assimilation* and *accommodation*. They form the basis for how information is incorporated into memory structure and how the structure adapts to ensure coherence.

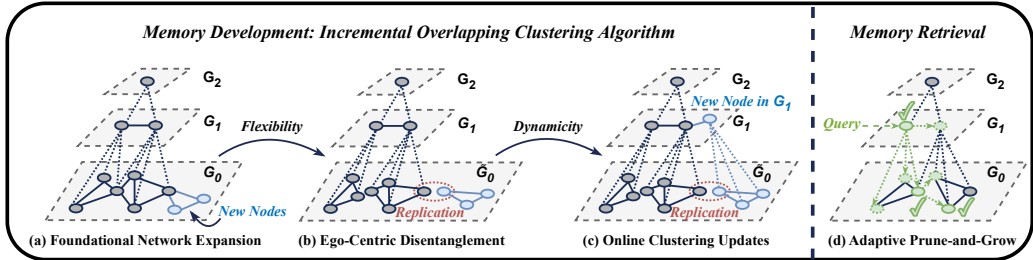

Figure 2: The memory development (Section 4.1) and retrieval (Section 4.2) processes of CAM.

## 3.2 Flexible Assimilation

Assimilation refers to the process of fitting information into the memory schemata without drastically altering its structure. Formally, given a batch of new information units $V_{new} = \{v_{n+1}, \ldots, v_{n+m}\}$, the memory system first organizes these units into the foundational semantic network $G_0$, updating it to $G_0' = (V \cup V_{new}, E \cup E_{new})$, where $E_{new}$ is the set of associative edges between the new units and current nodes, as well as among the new units themselves. This step ensures that the foundational network remains a comprehensive representation of the semantic landscape for all input information.

On top of $G_0'$, the next critical step of assimilation is to establish the hierarchical affiliations of each newly added node $v_{new} \in V_{new}$. Specifically, the memory system associates $v_{new}$ with the high-level abstractions in $G_1$ by determining the upward mapping $\psi_1(v_{new})$. This mapping can either assign each node to existing abstractions (i.e., $\psi_1(v_{new}) \subseteq V_1$) to enrich their semantic scope, or create entirely new abstraction nodes into $G_1$ to trigger further assimilation at higher levels of memory.

The key feature of assimilation lies in its *flexibility*, allowing each lower-level information unit to simultaneously enrich multiple higher-level abstractions, i.e., $\psi$ should be a *many-to-many* mapping. By enabling such overlapping affiliations, the memory schemata can capture the multifaceted nature of complex input information, where a single unit may relate to multiple themes, topics, or concepts.

## 3.3 Dynamic Accommodation

During assimilation, the memory schemata incrementally expand with the integration of new nodes and edges. Such expansions, however, may render the memory hierarchy suboptimal or misaligned with the updated context. For instance, an abstraction node may become overloaded, compromising its effectiveness in representing its subordinate units. In response, accommodation involves altering affected parts of the hierarchy, redistributing subordinate units and recalibrating abstraction nodes to restore structural coherence and cognitive equilibrium. This dynamic process maintains balanced memory schemata, efficiently adapting to new information without the need for global restructuring.

This trait of memory is particularly vital for processing long-form documents in real-world scenarios, where texts may arrive incrementally (e.g., in streaming or batch settings). For example, when a new chapter of a book is introduced, the memory system should integrate it by refining only the relevant parts of the schemata—such as updating a high-level abstraction representing the book's overarching narrative—rather than rebuilding the entire hierarchy from scratch. This capability ensures scalability and efficiency, as the memory system can adapt to new inputs in a continuous, online manner.

# 4 Prototype: Constructivist Agentic Memory

Based on the above blueprint, we then develop a prototype of Constructivist Agentic Memory (CAM) for LLM-based reading comprehension over long-form documents. This framework, illustrated in Figure 2, embodies the structurality, flexibility, and dynamicity in the construction of reading memory (Section 4.1), and adopts an adaptive inference strategy to handle user queries (Section 4.2).

## 4.1 Memory Development

In the reading phase, CAM is expected to flexibly assimilate and dynamically accommodate the input texts within a coherent hierarchy. Our technical implementation draws upon two observations. First, the flexible assimilation aligns well with *overlapping clustering*, where closely related memory nodes

form clusters and each node can belong to multiple clusters. Meanwhile, the dynamic accommodation corresponds to *incremental clustering*, where the cluster structures evolve in response to new inputs. These two cognitive processes are intertwined rather than independent, suggesting that a compact, unified algorithmic design integrating both functionalities would be preferable. Thus, we empower CAM with an incremental overlapping clustering algorithm for memory development, ensuring both simplicity and efficiency. Specifically, our procedure comprises three primary steps:

1. **Foundational Network Expansion:** New text chunks $V_{new}$ are integrated into the foundational semantic network $G_0 = (V, E)$ (initially empty), with the new edges $E_{new}$ established based on *textual relevance* and *narrative coherence*.

2. **Ego-Centric Disentanglement:** For the affected nodes $A$ (new nodes and their neighbor nodes), CAM analyzes their local structures to update the *replica network* $\tilde{G}_0$ (initially empty) of $G_0$, which explicitly disentangles the overlapping clusters through *node replication*.

3. **Online Clustering Updates:** On the non-overlapping replica network $\tilde{G}_0$, an incremental label propagation algorithm is applied to modify the cluster assignment. For modified clusters, CAM then updates the abstraction nodes using LLMs, and the affected supernodes further trigger this procedure (Step 2 and Step 3) at the next level.

In the following, we elaborate on these three steps in detail. For clarity, we focus on the development process from the foundational $G_0$ to the next $G_1$, which can be naturally extended to higher layers.

### 4.1.1 Foundational Network Expansion

Given $m$ new contiguous text chunks $V_{new} = \{v_{n+1}, v_{n+2}, ..., v_{n+m}\}$, CAM first incorporates these information units into the foundational semantic network $G_0 = (V, E)$, laying the groundwork for subsequent hierarchical development. To capture the textual relevance and narrative coherence among the received chunks, we define a composite score function that integrates both *semantic similarity* and *positional proximity*. Formally, for each pair of chunks $v_i \in V_{new}$ and $v_j \in V \cup V_{new}$, the semantic similarity is computed as the cosine similarity between their embeddings produced by a pretrained model $f_{emb}(\cdot)$. In addition, we measure the positional proximity between $v_i$ and $v_j$ using Gaussian similarity, which assigns higher scores to chunks that are closer in position. Combining these two aspects, the overall similarity score is defined as their linear interpolation:

$$s(v_i, v_j) = \alpha \cdot \frac{f_{emb}(v_i) \cdot f_{emb}(v_j)}{\|f_{emb}(v_i)\|\|f_{emb}(v_j)\|} + (1 - \alpha) \cdot \exp\left(-\frac{(i-j)^2}{2\sigma^2}\right), \qquad (2)$$

where $\alpha \in [0, 1]$ is the weighting coefficient, $i$ and $j$ are positional indices of $v_i$ and $v_j$, respectively, and $\sigma$ controls the decay rate of proximity influence. For each received chunk, we identify the top-$k$ relevant nodes whose similarity scores exceed a predefined threshold $\theta$, and then establish new edges between them to form the expanded foundational graph $G'_0 = (V', E') = (V \cup V_{new}, E \cup E_{new})$.

### 4.1.2 Ego-Centric Disentanglement

Memory assimilation allows each lower-level unit to simultaneously contribute to multiple higher-level abstractions. CAM achieves such flexibility through an ego-centric disentanglement strategy, which separates the contributions of each node by replicating it based on its local structures.

For every node $v \in V$ in $G_0$, CAM first extracts its ego-network $G_0[\mathcal{N}(v)]$, which is the subgraph consisting of the neighbor nodes of $v$ and the edges among them [39, 40].[1] From this local perspective, CAM then partitions $G_0[\mathcal{N}(v)]$ into connected components $\{C_v^1, C_v^2, \ldots, C_v^{t_v}\}$, where $t_v$ denotes the number of components in the ego-network of $v$. These components reflect the multifaceted roles of node $v$, which acts as the local cut point bridging otherwise disconnected contexts. To explicitly model such roles, $t_v$ replicas of $v$ are created, denoted as $\{v^1, v^2, ..., v^{t_v}\}$, each exclusively associated with one connected component. Let $\tilde{V}$ denote the set of replicas of all nodes in $G_0$, every original edge is also mapped to the connection between the replicas of its endpoints: if $(u, v) \in E$, $u \in C_v^j$, and $v \in C_u^i$, then an edge $(u_i, v_j)$ is added to $\tilde{E}$. The resulting network $\tilde{G}_0 = (\tilde{V}, \tilde{E})$ effectively disentangles the overlapping structures through node replication [40], thus allowing information to be summarized into multiple abstractions even with simple non-overlapping clustering algorithms.

---

[1]Note that the ego-network of $v$ does not include the node $v$ itself in this context [39].

Notably, this disentanglement process is naturally suitable for dynamic settings. When a set of new nodes $V_{new}$ and corresponding edges $E_{new}$ are integrated into $G_0$ (as described in Section 4.1.1), CAM only needs to construct or update the replicas of the affected nodes $A = V_{new} \cup \{u \in V | \exists v \in V_{new}, (u, v) \in E_{new}\}$, without the need for a full reconstruction. Moreover, since all nodes focus solely on their respective ego-networks, the replication of multiple nodes is inherently parallelizable.

### 4.1.3 Online Clustering Updates

On the replica network $\tilde{G}_0$, CAM then employs an incremental label propagation algorithm for online clustering updates. This procedure operates locally over a subgraph of $\tilde{G}_0$, thereby enabling efficient adaptation to new received information. Specifically, when a set of replicas is assimilated into $\tilde{G}_0$ (as described in Section 4.1.2), CAM first tracks all affected nodes $\tilde{A}$, including the newly added replicas and the existing replicas whose neighborhoods have changed. The new replica nodes are initialized with unique cluster labels, while the existing ones retain their previous labels. Then, CAM performs label propagation within the induced subgraph of $\tilde{A}$, where the label of each node $v \in \tilde{A}$ is iteratively updated according to the majority label among its neighbors $\mathcal{N}(v)$. If an existing node's cluster label changes, all its neighbors are added to $\tilde{A}$ for the next round of updates, ensuring that potential ripple effects are fully captured. This process continues recursively until no further label changes occur or a predefined iteration limit is reached. Note that we adopt such an algorithm for its scalability [41], and other incremental strategies [42] could also be used in this step depending on specific requirements.

Once the clustering process converges, CAM aggregates the nodes in each modified cluster to update their corresponding abstraction nodes in the next memory layer. This aggregation transforms tightly connected text chunks into compact, coherent summaries through LLM-based textual summarization. The resulting supernodes and their inter-cluster connections are then incorporated into the higher-level memory network as new nodes and edges, triggering further development of the memory hierarchy.

## 4.2 Memory Retrieval

Once the memory structure is constructed, another critical issue for reading agents is how to retrieve relevant information from the memory in response to user queries. Existing methods typically adopts two common strategies: hierarchical traversal [14, 35] and global retrieval [14, 18, 30] (Appendix B). However, they both have inherent limitations. Hierarchical traversal compares the input query to only a subset of nodes in each memory layer, making it prone to missing relevant information, especially in lower layers [14]. Global retrieval, while enabling comparisons across all nodes, simply relies on one-time semantic matching and overlooks the associated memory structures.

In this work, we empower CAM with a associative retrieval strategy, which first rapidly locates query-related cues from a global view, and then performs recursive associations along the memory structure. Specifically, our strategy proceeds in two stages:

1. **Fast Localization:** For an input query $q$, CAM first calculates the cosine similarity between the query embedding $f_{emb}(q)$ and the embedding of each memory node $f_{emb}(v)$. The top-$s$ nodes with the highest similarity scores are selected to form a candidate set $D$. This stage mirrors the human cognitive process of rapidly pinpointing relevant memory fragments during recall.
2. **Associative Exploration:** Then, CAM uses an LLM to select nodes from $D$ that are helpful for answering the query, forming an activation set $P \subseteq D$. Subsequently, the same-layer neighbors and lower-layer children of all activated nodes are collected into a new candidate set, from which the LLM continues to select potentially useful nodes to expand the activation set $P$. This process repeats until $P$ no longer grows or a maximum number of iterations is reached. Finally, all activated nodes are fed into the LLM for inference, akin to human associative thinking.

This *Prune-and-Grow* workflow combines global semantic matching and local structural exploration, enabling the LLM to adaptively gather query-relevant memories as support for contextual inference.

# 5 Experiments

To comprehensively validate the effectiveness of our CAM framework, we evaluate its performance on a range of single- and multi-document reading comprehension tasks, including question answering, query-based summarization, and claim verification. In the main experiment (Section 5.2), we adopt an

Table 2: Reading comprehension results. **ACC-L** is the accuracy score from the LLM judge. **R-1** and **R-L** are ROUGE F-Measures. The best results are in **bold**, and the second best results are underlined. Note that FullContext is inherently not suitable for the multi-document scenarios.

| Methods | Single-Document Scenarios | | | | | | | | Multi-Document Scenarios | | | | | | | |
| | NovelQA | | | QMSum | | | FABLES | | MH-RAG | | ODSum-Story | | | ODSum-Meeting | | |
| | R-1 | R-L | ACC-L | R-1 | R-L | ACC-L | F1$_P$ | F1$_N$ | EM | F1 | R-1 | R-L | ACC-L | R-1 | R-L | ACC-L |
|---|---|---|---|---|---|---|---|---|---|---|---|---|---|---|---|---|
| FullContext | 19.5 | 17.2 | 38.2 | 26.3 | 17.5 | 15.2 | 79.5 | 39.1 | - | - | - | - | - | - | - | - |
| MemGPT [11] | 21.3 | 19.5 | 40.8 | 31.4 | 20.2 | 40.3 | 81.3 | 38.5 | 63.2 | 67.1 | 32.6 | 18.4 | 36.8 | 28.5 | 16.5 | 33.2 |
| ReadAgent [13] | 22.1 | 20.4 | 42.3 | 32.9 | 23.6 | 45.5 | 84.2 | 43.6 | 60.8 | 65.5 | 33.5 | 19.0 | 39.4 | 28.8 | 16.3 | 35.7 |
| RAPTOR [14] | 26.5 | 23.7 | 47.8 | 34.2 | 24.5 | 50.7 | 86.5 | 48.5 | 69.4 | 73.6 | 37.4 | 24.0 | 48.7 | 32.6 | 19.9 | 44.3 |
| GraphRAG [15] | 24.8 | 22.5 | 45.3 | 35.8 | 25.2 | 53.9 | 87.2 | 48.0 | 65.2 | 70.3 | 37.2 | 24.3 | 50.2 | 33.7 | 20.5 | 45.8 |
| HippoRAG [16] | 25.5 | 23.0 | 46.5 | 31.9 | 20.8 | 41.2 | 84.7 | 45.8 | 67.9 | 72.5 | 34.5 | 22.8 | 42.9 | 30.2 | 18.3 | 41.2 |
| MemTree [18] | 23.4 | 21.2 | 43.5 | 33.7 | 22.9 | 48.3 | 83.1 | 46.6 | 66.5 | 71.2 | 35.2 | 23.2 | 46.0 | 31.8 | 19.5 | 42.5 |
| CAM | **28.8** | **25.4** | **52.3** | **37.2** | **26.5** | **57.6** | **91.5** | **52.5** | **72.8** | **77.5** | **39.2** | **25.5** | **54.6** | **35.9** | **23.2** | **50.7** |
| Δ (%) | **+2.3** | **+1.7** | **+4.5** | **+1.4** | **+1.3** | **+3.7** | **+4.3** | **+4.0** | **+3.4** | **+3.9** | **+1.8** | **+1.2** | **+4.4** | **+2.2** | **+2.7** | **+4.9** |

offline setting, allowing CAM to access all texts at once to build the memory. Then, we demonstrate the dynamicity of CAM under a batch-level online setting (Section 5.3), where CAM accesses the texts in batches. More analysis of configurations and variants is provided in Section 5.4.

## 5.1 Experimental Setup

**Datasets**    For single-document scenarios, we use NovelQA (question answering) [43], QMSum (summarization) [44], and FABLES (claim verification) [45]. For multi-document scenarios, we evaluate on MultiHop-RAG (question answering) [46], ODSum-Story (summarization) [47], and ODSum-Meeting (summarization) [47]. We select these datasets for their high quality, diversity, and long-text characteristics. See Appendix C for more details and statistics. We also demonstrate CAM's effectiveness on more classic datasets, with details provided in Appendix G due to space constraints.

**Baselines**    We compare CAM with two categories of baselines: (1) *non-structured memory*, including FullContext (a naive baseline that feeds full documents with truncation), MemGPT [11], and ReadAgent [13]; (2) *structured baselines*, including RAPTOR [14], GraphRAG [15], HippoRAG [16], and MemTree [18]. More descriptions of these baselines are provided in Appendix D.

**Metrics**    Considering the free-form reference responses in NovelQA, QMSum, and ODSum datasets, we report two widely-used metrics [13, 14, 18]: ROUGE F-Measures [48] for lexical similarity and LLM-as-a-judge Accuracy (ACC-L) from GPT-4o [49]. For MultiHop-RAG (MH-RAG) with definite short reference answers, we report the exact match (EM) and F1 scores. For FABLES, we follow its original paper [45] to report the separate F1 scores for positive and negative labels, i.e., F1$_P$ and F1$_N$.

**Implementation Details**    Unless otherwise specified, we use GPT-4o-mini as the LLM backbone and text-embedding-3-small as the embedding model $f_{emb}$. For fair comparisons, we standardize the use of LLMs and embedding models across all approaches to ensure that the observed performance differences are attributed to the memory design, rather than the variations in the LLMs or embeddings. The project code is available at https://github.com/rui9812/CAM. More implementation details can be found in Appendix E.

## 5.2 Offline Performance Comparison

Table 2 presents the main results on the six reading comprehension benchmarks. One can observe that CAM consistently outperforms the baselines across all metrics over these datasets. More insightfully, these results align well with the constructivist design principle in terms of structurality and flexibility.

**Memory Structurality**    Structured memory approaches (e.g., RAPTOR, GraphRAG, and our CAM) demonstrate clearly stronger performance against those without explicit structures (e.g., ReadAgent and MemGPT), confirming the importance of memory structurality in reading comprehension tasks. Hierarchical structures are particularly effective for summarization datasets (QMSum and ODSum), as evidenced by the superior results of RAPTOR, GraphRAG, and MemTree over the non-hierarchical HippoRAG. Notably, LLM-driven knowledge graph modeling in GraphRAG and HippoRAG does not bring consistent gains compared to RAPTOR, likely due to difficulty in extracting informative entities from narrative-centric texts. More analysis is provided in Section 5.4 and Appendix G.

**Assimilation Flexibility**    Despite both employing the tree-like memory structure and global retrieval, RAPTOR consistently surpasses MemTree across all datasets. This performance gap (averaging 2.1%

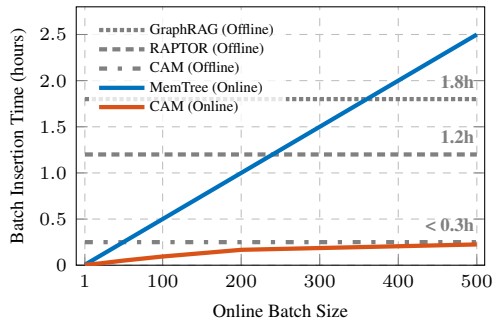
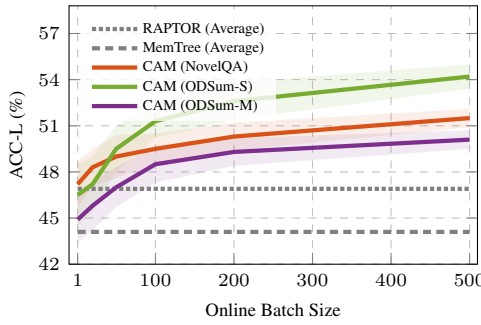

(a) Average Insertion Time vs. Online Batch Size  (b) ACC-L Performance vs. Online Batch Size

Figure 3: CAM's advantages in insertion efficiency and performance stability under the online setting.

on all metrics) directly stems from their difference in assimilation flexibility: RAPTOR permits each node to participate in multiple summaries, while MemTree enforces strict hierarchical containment. This comparison confirms that flexible assimilation is crucial for effective long-text comprehension.

**Memory Retrieval**    We also notice the performance variations between RAPTOR and GraphRAG on different tasks, underscoring the impact of memory retrieval strategies on the overall performance. RAPTOR's global retrieval is suitable for question answering, while GraphRAG integrates all memory nodes to respond, making it more advantageous in the comprehensive summarization tasks.

**CAM's Holistic Advantages**    Following the constructivist design principle, our CAM framework embodies both structurality and flexibility, with an adaptive prune-and-grow strategy to explore the memory hierarchy. In this way, CAM is not only suitable for the question answering tasks but also demonstrates significant advantages over the summarization tasks. Specifically, our CAM consistently achieves superior performance on all datasets, delivering an average gain of 3.0% across all metrics compared to the best baselines (RAPTOR and GraphRAG). Furthermore, we would like to highlight another unique advantage of CAM in the next part: its applicability to the batch-level online setting.

## 5.3    Dynamicity of CAM: From Offline to Online

Real-world application scenarios (e.g., reading serialized novels or tracking real-time news) typically require the memory systems to continuously integrate new texts in batches while maintaining stability. Existing offline methods (e.g., RAPTOR and GraphRAG) inherently demand full memory reconstruction for each update. MemTree, though online, only allows sequential, chunk-by-chunk integration. Unlike them, CAM naturally supports batch-level assimilation and local-first accommodation, thereby ensuring both processing efficiency and inference stability of the online memory development.

**Processing Efficiency**    Figure 3(a) shows the average integration time for a new batch of text chunks (each chunk contains 512 tokens) for different memory methods. Offline RAPTOR and GraphRAG require over 1 hour to rebuild entire memories, making them unsuitable for real-time use. MemTree, due to its sequential processing nature, incurs linearly increasing time with the batch size, even slower than offline reconstruction when the new batch exceeds 400 chunks. In contrast, our CAM maintains efficient integration across all batch sizes based on its parallelizable assimilation and localized accommodation processes. As the batch size increases, CAM's time cost grows sublinearly, approaching its offline level ($4\times$ faster than RAPTOR and GraphRAG), with LLM operations as the main overhead. Such a time convergence is reasonable, since large batches require substantial memory adjustment.

**Inference Stability**    Figure 3(b) reports the ACC-L results of CAM across different batch sizes on NovelQA, ODSum-Story, and ODSum-Meeting.[2] The performance of online CAM remains relatively stable across different batch sizes, and continues to be competitive against the baselines in Table 2. This indicates that CAM is able to effectively adjust its memory structure to accommodate new texts.

To conclude, CAM provides an efficient and reliable solution for long-context reading comprehension in online scenarios. The designed assimilation and accommodation processes effectively address the critical demands of processing efficiency and inference stability, establishing CAM as the first agentic memory framework that simultaneously possesses structurality, flexibility, and dynamicity.

---

[2]We use these datasets as they contain extremely long documents (Appendix C) and share the same metrics.

Table 3: Dissecting CAM with different configurations (rows 2-5) and variants (rows 6-10).

| | | NovelQA | | QMSum | | MH-RAG | | ODSum-S | |
|---|---|---|---|---|---|---|---|---|---|
| | | R-L | ACC-L | R-L | ACC-L | EM | F1 | R-L | ACC-L |
| *CAM with GPT-4o-mini and text-embedding-3-small* | | 25.4 | 52.3 | 26.5 | 57.6 | 72.8 | 77.5 | 25.5 | 54.6 |
| **LLM Backbones** | Llama-3.1-8B-Instruct | 23.7 | 49.5 | 25.3 | 54.4 | 70.3 | 74.2 | 24.3 | 52.5 |
| | Qwen2.5-7B-Instruct | 24.1 | 50.6 | 25.5 | 55.0 | 70.9 | 75.6 | 24.2 | 53.2 |
| **Embedding Models** | text-embedding-3-large | 25.8 | 53.0 | 27.4 | 59.2 | 73.2 | 78.0 | 26.7 | 56.4 |
| | E5-Mistral-7B-Instruct | 26.4 | 54.3 | 27.1 | 58.7 | 73.5 | 78.4 | 26.3 | 55.7 |
| **Retrieval Strategy** | Hierarchical Traversal | 23.2 | 46.5 | 24.8 | 55.2 | 68.3 | 72.5 | 23.8 | 49.5 |
| | Global Retrieval | 24.5 | 50.3 | 25.7 | 56.3 | 70.5 | 75.9 | 24.4 | 50.8 |
| **Variants** | w/ Fine-Grained Modeling | 25.8 | 53.5 | 26.3 | 57.2 | 72.0 | 76.8 | 25.1 | 54.8 |
| | w/o Hierarchy | 22.9 | 46.7 | 23.2 | 46.6 | 68.5 | 73.4 | 23.5 | 49.3 |
| | w/o Flexibility | 24.2 | 50.3 | 24.7 | 51.5 | 70.3 | 75.1 | 24.7 | 51.2 |

## 5.4 More Analysis: Configurations and Variants

**LLM Backbones**    To assess whether CAM's performance relies on commercial LLMs like GPT-4o-mini, we replace it with two popular open-source LLMs: Llama-3.1-8B-Instruct and Qwen2.5-7B-Instruct. As shown in Table 3 (rows 2-3), despite their relatively smaller parameter scales, the CAM implementations based on these two LLMs still deliver strong performance, with only a slight drop compared to the GPT-4o-mini version (row 1). This suggests that CAM's core strengths primarily stem from the effective memory design, rather than relying on the expensive LLM backbones.

**Embedding Models**    We further study the impact of embedding models on our CAM's performance. Beyond the default text-embedding-3-small, we employ text-embedding-3-large and E5-Mistral-7B-Instruct, two stronger embedding models on the MTEB leaderboard [50]. Table 3 (rows 4-5) shows that the use of more advanced embedding models can improve the performance of CAM, since they deliver more accurate similarity measurements for the construction of memory foundational networks.

**Retrieval Strategies**    Table 3 (rows 6-7) reports the results of CAM under different memory retrieval strategies. We observe that using either hierarchical traversal or global retrieval leads to performance degradation, validating the effectiveness of our Prune-and-Grow strategy. Notably, CAM centers on the memory development process and is thus inherently compatible with diverse retrieval strategies, allowing users to select the suitable one based on their specific needs in real-world applications [51].

**Performance by Question Type**    To further investigate how CAM performs across varying question characteristics, we conduct an in-depth performance analysis on the NovelQA dataset by leveraging its available question-type annotations. Specifically, the questions are categorized along two orthogonal axes (i.e., *complexity-based* and *aspect-based* categories) to reflect their reasoning complexity and semantic focus. Detailed results are provided in Appendix F. We observe that CAM exhibits clear advantages in handling the complex questions (e.g., *multi-hop*, *times*, and *span* types), which typically require integrating semantically distant evidence and maintaining narrative coherence.

**Fine-Grained Foundational Modeling**    Several recent works (e.g., GraphRAG and HippoRAG) employs LLMs to construct a knowledge graph from input texts (i.e., entity recognition and relation extraction). Such fine-grained knowledge modeling can be readily integrated into CAM for building a more granular foundational network. However, this process would increase the computational cost by more than threefold due to the extensive LLM invocations, yet it does not achieve commensurate improvements as shown in Table 3 (row 8). We observe that even commercial LLMs (GPT-4o-mini) struggle to extract informative entities from lengthy narratives. See Appendix G for more analysis.

**Ablation Variants**    Table 3 (rows 9-10) presents the performance of two ablation variants of CAM. One variant omits hierarchical clustering entirely, relying solely on the foundational semantic network for inference; while the other bypasses the ego-centric disentanglement and directly applies the label propagation for hierarchical clustering. One can see that both variants result in clear performance degradation, thereby confirming the importance of hierarchy and flexibility in our design principle.

## 6 Limitations and Discussion

**Beyond Reading Comprehension**    While constructivist theory offers broad insights into cognitive development, our work is tailored specifically to long-text reading comprehension tasks, such as question answering, query-based summarization, and claim verification. Extending such a profound

memory design principle to other domains, such as behavioral planning, long-sequence generation, and multi-modal tasks, remains unexplored but holds significant potential for future investigation.

**More Agentic Behaviors**   This work primarily focuses on delineating the key traits of agentic memory and develops a prototype to validate the importance of these traits. However, additional agentic behaviors, such as self-questioning and reflection, are not incorporated into our framework. Integrating these capabilities could be crucial for advancing more robust LLM-based agent systems.

**Hallucination Propagation**   CAM leverages LLMs for summarization during memory development, posing the risk of hallucination—generating inaccurate or fabricated information. As CAM generates hierarchical summaries, errors or hallucinations in lower-level nodes may propagate to higher-level abstractions, potentially affecting the applicability of our framework in real-world scenarios. Detecting and mitigating such hallucinations within the agentic memory remains an open challenge.

**Inconsistent Information Sources**   The ability to detect and reconcile contradictory information is a critical aspect of human cognition. In line with prior LLM-based memory systems (e.g., GraphRAG, RAPTOR, and MemTree), our framework also assumes that the source texts are internally consistent. However, real-world documents often contain conflicting facts or perspectives, especially in complex open-domain settings. Addressing this challenge requires dedicated efforts, such as constructing new benchmarks and designing evaluation protocols to measure the reconciliation capabilities. We consider the inconsistency-aware memory development as a promising direction for future research.

**Alternative Implementations**   CAM instantiates the constructivist design principle with a local-first incremental overlapping clustering algorithm. While this implementation fulfills the desired properties (i.e., structured schemata, flexible assimilation, and dynamic accommodation), it is not the only path toward such goals. Other strategies (e.g., neural controllers and symbolic planners) may offer different trade-offs in scalability, interpretability, and generalizability. Exploring alternative implementations that adhere to the constructivist principle could further enrich the landscape of memory systems.

**Learn to Memorize**   Our CAM implementation relies on fixed prompts and tuned hyperparameters rather than adaptive policies for memory assimilation and accommodation. It neither optimizes the memory structure nor adapts its update rules from downstream feedback. Moving toward trainable memory controllers (e.g., learning what to update or how to route retrieval) could further enhance the quality and efficiency of memory development. Designing such mechanisms poses non-trivial challenges in credit assignment, but marks a key step toward more powerful agentic memory systems.

## 7   Conclusion

This paper focuses on an emerging problem: how to design the memory modules for LLMs to build reading agents capable of comprehending extremely long-form documents. To address this, we draw upon Jean Piaget's Constructivist Theory, positing that an effective memory module should possess *structured schemata*, *flexible assimilation*, and *dynamic accommodation*. Building on this blueprint, we develop a prototype of Constructivist Agentic Memory (CAM), which shows dual advantages in both performance and efficiency across a wide range of long-text reading comprehension benchmarks. We hope that our proposal will inspire broader exploration of cognitive memory architectures to build more powerful LLM agents. Moreover, we believe that the constructivist design principle also offers a compelling foundation for advancing agentic memory beyond textual understanding, with potential applications in planning, reasoning, and multi-modal domains.

## Acknowledgments and Disclosure of Funding

This work is supported in part by National Natural Science Foundation of China (No. 62422215 and No. 62472427), Major Innovation & Planning Interdisciplinary Platform for the "DoubleFirst Class" Initiative, Renmin University of China, Public Computing Cloud, Renmin University of China, fund for building world-class universities (disciplines) of Renmin University of China, the Outstanding Innovative Talents Cultivation Funded Programs 2024 of Renmin University of China, and Huawei Innovation Research Programs. We gratefully acknowledge the support from Mindspore[3], CANN (Compute Architecture for Neural Networks) and Ascend AI Processor used for this research.

---

[3]`https://www.mindspore.cn`

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

# A  Overview of Jean Piaget's Constructivist Theory

Jean Piaget's Constructivist Theory [19–21], originally formulated to explain cognitive development in children, offers a valuable framework for understanding how readers comprehend complex texts. This theory emphasizes that reading comprehension is not solely a passive reception of information, but involves an active, constructive process. Central to this process are the concepts of schemata, assimilation, and accommodation, which work together to drive cognitive development.

**Schemata**   Schemata are cognitive structures or mental frameworks that organize and interpret information, serving as the building blocks of cognition. In reading comprehension, this layered organization enables readers to process information at different levels of granularity, from basic word recognition to higher-order conceptual analysis. For example, a child's basic schema for "animal" may include broad traits like "moves" or "eats", which later differentiates into more specific schemata for "dog" or "bird". These hierarchical structures provide stability, enabling efficient processing of familiar experiences, while their layered organization supports the integration of increasingly abstract and sophisticated knowledge.

**Assimilation**   Assimilation is the process of integrating new experiences into existing schemata with minimal adjustment. It reflects cognitive flexibility, allowing individuals to apply familiar frameworks to new situations. For example, a child encountering a new breed of dog might assimilate it into their "dog" schema by recognizing shared features like fur or barking. Assimilation is essential for learning, as it builds on the hierarchical structure of schemata, reinforcing and expanding existing knowledge without disrupting cognitive organization. However, when new experiences challenge these frameworks, a complementary process is required.

**Accommodation**   Accommodation involves modifying existing schemata or creating new ones to account for experiences that do not fit current frameworks. This dynamic process drives cognitive growth by restructuring the hierarchical organization of schemata to align with reality. For example, if a child mistakes a cat for a dog, they may accommodate by creating a new "cat" schema or refining their "dog" schema to exclude cats. Accommodation is vital for resolving cognitive conflicts, enabling the development of more accurate and complex schemata within the hierarchical system.

**The Interplay of Assimilation and Accommodation**   Piaget's concept of equilibration describes the dynamic interplay between assimilation and accommodation, which maintains cognitive balance. When new experiences cause disequilibrium, individuals use assimilation to incorporate compatible information and accommodation to adjust or expand their hierarchical schemata. This cycle drives progression through Piaget's stages of development (sensorimotor, preoperational, concrete operational, and formal operational), as schemata become increasingly differentiated and organized.

In summary, Piaget's constructivist theory emphasizes the hierarchical organization of schemata as the foundation of knowledge, with flexible assimilation and dynamic accommodation enabling adaptation and growth. These processes work together to refine and expand cognitive hierarchies, making Piaget's theory a powerful lens for understanding memory development.

# B  Discussion on Related Works

Table 1 summarizes the traits of current representative memory works from the constructivist perspective. Here, we provide a detailed exposition of these methods.

**Tabular Memory**   Unstructured methods, such as MemGPT [11] and ReadAgent [13], store or compress individual text chunks independently into tabular memory. Due to their design simplicity, these methods naturally support batch processing of new texts. However, they disregard the significance of memory structurality, thus rendering discussions of flexibility and dynamicity irrelevant.

**RAPTOR [14]**   This method recognizes the importance of higher-level summarization, adopting a bottom-up clustering and summarization strategy to form hierarchical memory, aligning with the concept of structured schemata. RAPTOR employs a soft clustering approach based on Gaussian Mixture Models, where nodes can belong to multiple clusters without requiring a fixed number of clusters, thereby fulfilling the flexibility of memory assimilation. However, the method operates in a fully offline manner, lacking incremental updates to memory, i.e., there is no dynamic accommodation. Moreover, the clustering method relies on Gaussian assumptions, which may not perfectly match the nature of text data, potentially affecting its applicability across different text types.

**GraphRAG [15] and HippoRAG [16]**   GraphRAG leverages the linguistic capabilities of LLMs to extract fine-grained knowledge triplets from each text chunk and construct a knowledge graph. Then, it employs the Leiden algorithm [52] for community detection and generates summaries with LLMs. This approach satisfies structurality but lacks flexibility, since the Leiden algorithm produces disjoint community structures such that each lower-level node can only belong to a single higher-level node. Moreover, GraphRAG operates in an offline manner, i.e., it needs to rebuild the entire memory when new texts arrive. HippoRAG similarly constructs a knowledge graph memory from long documents but neglects the importance of higher-level semantic summaries. In our experiment, we observe that such fine-grained knowledge modeling remains challenging for LLMs, primarily due to the difficulty in maintaining the consistency of entities and relations over long texts. Further analysis in Appendix G reveals that such knowledge modeling is more suitable for factual QA than for narrative QA.

**MemTree [18]**   MemTree is the first method to consider the online scenarios, employing a top-down hierarchical clustering algorithm for memory development. However, it processes new input texts in a chunk-by-chunk manner, with only a single chunk inserted into the memory structure at a time. This design leads to inefficiency when handling a large number of chunks, as shown in Figure 3(a). MemTree assigns each chunk to a single position, thereby lacking flexibility. Furthermore, it does not adjust the memory structure to accommodate each new insertion, which may result in an imbalanced memory structure and sensitivity to the order of chunk insertion.

**Other related works**   Some other studies [53–56] also leverage structured approaches to enhance LLMs. SG-Prompt [53] first constructs a semantic graph structures through information extraction from all retrieved texts, and then leverages this symbolic information to enhance the inference quality of LLMs. GE-Reasoning [55] and Semi-Structured CoT [57] focus on parsing input questions into masked structured chains and subsequently fill each incomplete knowledge triplet based on a pre-defined KG or a plain text database. CoK [58] retrieves knowledge triplets from a pre-defined KG and then combines them with human annotations, aiming to construct rational exemplars to elicit the knowledge generation capabilities of LLMs. Most of these methods can be viewed as variants of GraphRAG, thus we do not include comparisons with them considering the evaluation budget.

In summary, none of these methods simultaneously satisfy the structurality, flexibility, and dynamicity requirements of constructivist theory. Our CAM framework effectively fills this void, offering an efficient and reliable solution for long-context reading comprehension in batch-level online scenarios.

## C   Dataset Statistics

We evaluate the performance of CAM on a range of single- and multi-document reading comprehension tasks, including question answering, query-based summarization, and claim verification.

For single-document scenarios, we use (1) **NovelQA** [43], a long-form novel question answering dataset, with an average of more than 200K tokens per novel; (2) **QMSum** [44], a query-based multi-domain meeting summarization dataset, with each meeting transcript averaging around 10K words; and (3) **FABLES** [45], a claim verification dataset that requires the model to judge the faithfulness of given claims on long-form documents, with an average of more than 120K tokens per document.

For multi-document scenarios, we further use (4) **MultiHop-RAG** [46], a recent multi-hop question answering dataset spanning a corpus of 609 news articles across six categories, with an average length of around 2,000 tokens per article; and (5) **ODSum** [47], a multi-document query-based summarization dataset consisting of two subsets (ODSum-Story and ODSum-Meeting).

For FABLES and MultiHop-RAG, we filter the unanswerable claims and questions. The statistics of all datasets are provided in Table 4.

Table 4: Statistics of six reading comprehesion benchmarks.

| Dataset Statistics | Single-Document Reading Comprehension | | | Multi-Document Reading Comprehension | | |
|---|---|---|---|---|---|---|
| | NovelQA | FABLES | QMSum | MultiHop-RAG | ODSum-Story | ODSum-Meeting |
| Task | QA | Verification | Summarization | QA | Summarization | Summarization |
| #Document | 89 | 26 | 232 | 609 | 1190 | 232 |
| #Token | 200K | 121K | 10K | 2046*609 | 809*1190 | 7176*232 |
| #Query | 2305 | 3158 | 1808 | 2255 | 635 | 436 |

# D  Baseline Descriptions

We compare CAM with a range of LLM-based reading comprehension approaches: (1) **FullContext**, a naive baseline that directly feed full documents into the LLM and truncates any content exceeding the context window; (2) **MemGPT** [10, 11], which retrieves query-relevant text chunks directly without formatting high-level representations; (3) **ReadAgent** [13], which generates summaries for all text chunks to compress the long documents into the LLM's context window; (4) **RAPTOR** [14], which recursively builds a tree structure with different levels of summarization; (5) **GraphRAG** [15], which extracts a knowledge graph from the documents and splits it into independent communities to form summaries; (6) **HippoRAG** [16], which similarly organizes a knowledge graph but lacks high-level abstractions; (7) **MemTree** [18], which sequentially integrates the texts into a tree from the top down and updates the summaries accordingly. More descriptions of these baselines and how they differ from our design are provided in Appendix B.

# E  Implementation Details

**Backbone models**    In our experiments, we standardize the use of LLMs and embedding models across all approaches to ensure that the observed performance differences are attributed to the memory design, rather than the variations in LLMs' capabilities or embeddings. By default, we use `GPT-4o-mini` with temperature of $0$ as the LLM backbone, and employ `text-embedding-3-small` as the embedding model $f_{emb}$. Different model configurations are also considered to demonstrate the applicability of our framework.

**Hyperparameter Selection**    Our CAM framework involves several key hyperparameters that control the memory development. We select the weighting coefficient $\alpha$ from $\{0.5, 0.7, 0.9\}$, the decay rate $\sigma$ from $\{1.0, 1.5, 2.0\}$, and the similarity threshold $\theta$ from $\{0.5, 0.7\}$. The chunk size is set to either $256$ or $512$ tokens, depending on the dataset characteristics. For memory expansion, we set $k$ to $10$, and for memory retrieval, we set $s$ to $5$. The temperature of the LLM is set to $0$. Note that for chunks not within in the same document, the proximity similarity in Equation (2) is $0$. All LLM prompts are listed in our code repository. As discussed in Section 6, while these hyperparameters are commonly used and empirically effective, they remain manually configured in CAM. A promising direction for future research is to develop automated mechanisms (e.g., reinforcement learning) to reduce reliance on manual tuning and enhance adaptability across diverse tasks.

**Memory Scope**    For single-document reading comprehension datasets (i.e., NovelQA, QMSum, and FABLES), CAM builds an independent memory structure for each document to facilitate reasoning over the entire document. For multi-document reading comprehension datasets (i.e., MultiHop-RAG, ODSum-Story, and ODSum-Meeting), a unified memory representation is constructed across all the involved documents, enabling CAM to retrieve and integrate information from multiple documents when responding complex queries.

**Metrics**    Considering the free-form reference responses in NovelQA, QMSum, and ODSum, we use two widely-used metrics for evaluation [13, 14, 18]: (1) ROUGE F-Measures [48], which capture the lexical similarity between the model output and the reference response; and (2) LLM-as-a-judge [49], which uses `GPT-4o` to compare the output with the reference, resulting in an accuracy score (ACC-L). For the factual question answering datasets MultiHop-RAG and HotpotQA, we use exact match (EM) and F1 scores to measure the performance. On the claim verification dataset FABLES, we follow its original paper [45] to report the separate F1 scores for positive and negative labels, i.e., $F1_P$ and $F1_N$.

**Compute Resource**    In our analysis experiment (Table 3), We run the open-source LLMs ( Llama-3.1-8B-Instruct and Qwen2.5-7B-Instruct) on an NVIDIA A100 GPU.

# F  Performance by Question Type

To provide a more nuanced evaluation of CAM, we also conduct a fine-grained performance analysis on the NovelQA dataset by leveraging its question-type annotations. These annotations enable us to investigate model behaviors along two orthogonal axes: (1) *complexity-based categories*, including *single-hop* (42.8%), *multi-hop* (35.0%), and *detail* (22.2%) questions, which reflect the reasoning depth required to arrive at an answer; and (2) *aspect-based categories*, covering *times*, *meaning*, *span*, *setting*, *relation*, *character*, and *plot*, which reflect the semantic focus of each query.

Table 5: Evaluation results on complexity-based categories.

| Methods | Single-Hop | Multi-Hop | Detail | Average |
|---------|-----------|-----------|--------|---------|
| ReadAgent | 45.3 | 39.5 | 41.0 | 42.3 |
| RAPTOR | 49.1 | 46.3 | 47.6 | 47.8 |
| GraphRAG | 47.7 | 44.2 | 42.5 | 45.3 |
| MemTree | 45.8 | 41.5 | 42.3 | 43.5 |
| CAM | 53.4 | 51.3 | 51.8 | 52.3 |

Table 6: Evaluation results on aspect-based categories.

| Methods | Times | Meaning | Span | Setting | Relation | Character | Plot | Average |
|---------|-------|---------|------|---------|----------|-----------|------|---------|
| ReadAgent | 32.5 | 31.7 | 30.3 | 48.2 | 43.5 | 49.1 | 49.5 | 42.3 |
| RAPTOR | 39.7 | 36.2 | 33.5 | 53.3 | 47.8 | 53.9 | 55.3 | 47.8 |
| GraphRAG | 37.1 | 33.4 | 32.8 | 50.5 | 52.0 | 48.6 | 53.4 | 45.3 |
| MemTree | 30.8 | 33.0 | 31.2 | 49.3 | 49.5 | 50.3 | 51.7 | 43.5 |
| CAM | 45.2 | 41.5 | 39.3 | 58.5 | 51.2 | 57.6 | 59.1 | 52.3 |

We compare CAM with four representative baselines (ReadAgent, RAPTOR, GraphRAG, MemTree) under identical settings. As shown in Tables 5 and 6, CAM consistently outperforms the baselines across these question categories, with particularly notable gains on complex questions that require integrating semantically distant evidence and maintaining contextual coherence. These results confirm CAM's superior generalization across varied reasoning depths and semantic focuses, underscoring its effectiveness in handling context-intensive reasoning tasks.

## G   Factual QA Performance

To verify the effectiveness of CAM more comprehensively, we further conduct experiments on three widely-used QA benchmarks: HotpotQA [59], MuSiQue [60], and 2WikiMultiHopQA (2Wiki) [61].

For their text corpora, we follow HippoRAG [16] and IRCoT [62] to collect all candidate passages (including supporting and distractor passages) from each dataset. We use the same model configuration (GPT-4o-mini and text-embedding-3-small) as in the main experiment to build the memory structures. For the evaluation metrics, we report the standard exact match (EM) and F1 scores.

Table 7: Evaluation results on three factual QA datasets.

| Methods | HotpotQA | | 2Wiki | | MuSiQue | |
|---------|------|------|------|------|------|------|
| | EM | F1 | EM | F1 | EM | F1 |
| RAPTOR | 59.2 | 73.4 | 58.4 | 66.2 | 23.3 | 34.8 |
| HippoRAG | 58.5 | 73.1 | 63.8 | 71.0 | 21.8 | 32.2 |
| CAM | 60.7 | 75.4 | 61.4 | 70.2 | 24.6 | 37.2 |
| CAM w/ FM | 62.1 | 76.5 | 65.8 | 75.3 | 26.8 | 38.5 |

Table 7 reports the performance of CAM and two representative baselines. Our framework consistently outperforms the baselines on HotpotQA and MuSiQue, while achieving competitive results on 2Wiki. We observe that HippoRAG exhibits notable performance across these datasets, particularly on 2Wiki. This may be attributed to its use of LLM-based fine-grained knowledge modeling, which is especially effective for entity-centric datasets [16]. To explore this, we further extend CAM with the fine-grained knowledge modeling and investigate its impact. As shown in Table 7, such an extension substantially enhances CAM's performance on both 2Wiki and MuSiQue. Moreover, we also manually inspect the quality of entities extracted by the LLM on 2Wiki and find that: compared to the narrative documents in our main experiments, the LLM is more capable of identifying informative entities and relations in entity-centric texts. This suggests that the fine-grained knowledge modeling technique is particularly suitable for factual knowledge QA tasks grounded in entity-rich contexts.

