# OpenReview forum: "CAM: A Constructivist View of Agentic Memory for LLM-Based Reading Comprehension"
_NeurIPS.cc/2025/Conference — NeurIPS 2025 poster_

### Official Review · Reviewer_vP7d · 2025-06-25

**Clarity:** 4
**Significance:** 3
**Originality:** 3
**Rating:** 4
**Confidence:** 4

**Summary:**

This paper introduces CAM (Constructivist Agentic Memory), a memory module for large language models (LLMs) inspired by Jean Piaget’s constructivist theory. CAM is guided by three key traits—structured schemata, flexible assimilation, and dynamic accommodation—and implements them via an incremental overlapping clustering algorithm for hierarchical memory construction and a prune-and-grow associative retrieval strategy. Empirically, CAM consistently outperforms a range of baselines on six long-text reading comprehension benchmarks (including question answering, query-based summarization, and claim verification), exhibiting both higher accuracy and substantially faster batch‐level memory updates.

**Questions:**

- The "Ego-Centric Disentanglement" step replicates nodes to achieve flexibility. Could you elaborate on the downstream effects of this? Does it lead to redundancy in the final retrieved context, and does managing a larger replica graph (Ğ₀) impact retrieval efficiency or memory footprint?
- The paper criticizes MemTree for its sensitivity to chunk insertion order. While CAM's batch processing mitigates this, could order effects still exist between batches? For instance, would processing batches in the order [A, B] then [C, D] yield a significantly different memory structure and performance compared to processing them in the order [D, C] then [B, A]? An analysis of this would further bolster the claims of dynamic robustness.
- While the paper states the framework is robust to its hyperparameters (θ, σ, α), a quantitative sensitivity analysis would be more convincing. For example, showing how key performance metrics vary across a range of values for these parameters would provide a clearer picture of the system's stability.

**Ethical Concerns:**

["NO or VERY MINOR ethics concerns only"]

**Final Justification:**

After considering the rebuttal and the additional experiments provided by the authors, I find that several of my original concerns were well addressed:
- The new retrieval efficiency comparisons, batch‐order robustness evaluation, and hyperparameter sensitivity analysis strengthened the empirical evidence for CAM’s claimed advantages.
- Clarifications on the “Ego‐Centric Disentanglement” mechanism and its impact on redundancy and efficiency resolved my earlier uncertainty in this area.

However, some issues remain unresolved within the current submission:
- The handling of contradictory information remains as future work rather than being addressed in the current implementation.
- While the authors acknowledge the risk of hallucination propagation, mitigation strategies are only discussed conceptually and not demonstrated experimentally.

Overall, the rebuttal improves my confidence in the method’s soundness and completeness, but given the remaining limitations, I maintain my original score.

**Limitations:**

Yes. The paper discusses its scope (limited to reading comprehension), omission of additional agentic behaviors (e.g., self-questioning/reflection), and the risk of hallucination propagation, all in Appendix I.

**Quality:**

3

**Strengths And Weaknesses:**

Strengths:
- Grounding the design in Piaget’s constructivist theory offers a clear, systematic framework, with structurality, flexibility, and dynamicity guiding both analysis and future designs.
- The incremental overlapping clustering, especially the “Ego‑Centric Disentanglement” step, neatly balances adaptation and growth, addressing rigidity and offline limits of earlier methods.
- Broad tests across tasks, document types, and settings—alongside strong baselines and ablations—support the method’s efficiency and stability, especially in online use.

Weaknesses:
- Relying on an LLM to summarize each cluster risks introducing mistakes or omissions at every layer. Without a dedicated correction mechanism, these errors can accumulate and distort higher‑level abstractions.
- The “Prune‑and‑Grow” strategy’s iterative LLM_Selection steps may incur substantial latency and computational expense during inference. While memory construction is shown to be efficient, retrieval efficiency isn’t thoroughly compared against baselines.
- Incorporating positional bias helps maintain narrative flow but may underweight semantically related segments that are spatially distant, potentially reducing recall for multi‑hop reasoning or thematic summaries.
- The framework assumes consistent source information and does not address how to detect or reconcile contradictory content within or across documents.

---

> ### Author Rebuttal · Authors · 2025-07-30
>
> Thanks for your careful and insightful comments. Below, we provide point-by-point responses to each concern raised.
> >### **Q1: Potential LLM summarization drift.**
>
> Thank you for raising this point. Using LLM to generate summaries has become a common practice in current agentic memory systems (e.g., ReadAgent, RAPTOR, GraphRAG, and MemTree). As we discussed in the Limitation section (**Appendix I**), we acknowledge that LLM-generated summaries may carry risks of hallucination propagation, especially in deeper memory layers. This limitation could be mitigated by introducing more agentic behaviors (such as self-questioning and reflection) to reduce the hallucination issue and improve the summary fidelity, which we believe is a promising direction for future memory research.
> >### **Q2: Inference-time latency from "Prune‑and‑Grow" retrieval strategy.**
>
> To investigate this point, we compare CAM's Prune-and-Grow retrieval with two common retrieval strategies (used in the baseline methods): Hierarchical Traversal (HT) and Global Retrieval (GR). To fully address your concern, we also isolate the impact of LLM selection by evaluating CAM with two non-LLM graph traversal strategies: Breadth-First Search (BFS) and Personalized PageRank (PPR). We report the inference performance (F1 for MH-RAG, R-L for others) / average time (in seconds) below.
>
> |Retrieval Methods|NovelQA|QMSum|MH-RAG|ODSum-Story|
> |:-:|:-:|:-:|:-:|:-:|
> |Prune-and-Grow|**25.4** / 7|**26.5** / 5|**77.5** / 8|**25.5** / 7|
> |HT|23.2 / 8|24.8 / 5|72.5 / 10|23.8 / 9|
> |GR|24.5 / 2|25.7 / 1|75.9 / 2|24.4 / 2|
> |BFS|23.3 / 3|24.7 / 2|75.4 / 3|23.5 / 3|
> |PPR|24.0 / 4|25.2 / 3|76.2 / 5|23.9 / 4|
>
> We observe that while the LLM-based retrieval incurs moderate overhead in response time, it brings consistently favorable performance gains beyond the baselines and non-LLM variants. The Prune-and-Grow process can effectively filter irrelevant memory nodes, resulting in more focused activation sets and improved downstream generation. Once the activation set stabilizes, the memory exploration then stop early to avoid unnecessary computation.
>
> Moreover, we would like to clarify that **CAM is not tied to a specific retrieval strategy**; rather, **it is inherently compatible with various retrieval methods**, allowing users to choose the most suitable one based on their latency and performance requirements in real-world applications.
> >### **Q3: Trade-off between positional bias and semantic relevance.**
>
> This balance is explicitly controlled by the **hyperparameter $\alpha$** in the similarity scoring function (Equation 2). When $\alpha = 1$, the influence of positional bias is fully removed, and chunk relevance is computed purely based on semantic similarity, which would facilitate the retrieval of spatially distant yet topically related segments.
>
> Incorporating positional bias helps maintain discourse coherence and improves retrieval of local context, aligning with human reading behavior where adjacent sentences are semantically cohesive. To quantify this effect, we conduct **a sensitivity analysis across different $\alpha$ values**, summarized below:
>
> |$\alpha$|NovelQA (R-L)|QMSum (R-L)|MH-RAG (F1)|ODSum-Story (R-L)|
> |:-:|:-:|:-:|:-:|:-:|
> |0.5|**26.1**|25.3|74.7|25.3|
> |0.7|25.4|26.5|77.5|**25.5**|
> |0.9|23.5|**26.7**|**78.3**|24.7|
> |1.0|23.2|26.2|77.9|24.5|
>
> To more thoroughly address your concern about hyperparameter robustness, we also provide a broader sensitivity analysis in our response to your **Q7**.
> >### **Q4: Handling of contradictory information.**
>
> We agree that the ability to detect and reconcile conflicting or contradictory information is a critical aspect of human cognition. Consistent with prior LLM-based memory systems such as GraphRAG, RAPTOR, and MemTree, our current framework assumes that source information is generally consistent and focuses on the memory development process. We consider memory reconciliation as a highly promising direction for future memory research, and promise to include a clear discussion of this point in our final version.
> >### **Q5: Discussion on the effects of "Ego-Centric Disentanglement".**
>
> **Effects.** Ego-Centric Disentanglement is central to CAM’s ability to form coherent and nuanced memory abstractions. Its core motivation lies in capturing the semantic multifacetedness of complex text chunks---each chunk often relates to multiple themes, topics, or concepts. By replicating nodes based on their local semantic neighborhoods, CAM allows a single chunk to contribute to multiple higher-level abstractions, enabling more comprehensive and context-sensitive summarization.
>
> **Does it lead to redundancy in the final retrieved context?** No. At inference time, chunk-level duplicates are automatically de-duplicated (based on unique chunk IDs). While multiple higher-level nodes may reference the same original chunk, these abstract nodes originate from distinct semantic clusters, making them meaningfully differentiated. To quantify this, we compute **pairwise semantic similarity** among all node embeddings in the memory structure, and the average results demonstrate low semantic redundancy:
>
> ||NovelQA|QMSum|MH-RAG|ODSum-Story|
> |:-:|:-:|:-:|:-:|:-:|
> |_Avg. Pair Similarity_|0.27|0.32|0.18|0.21|
>
> **Does managing a replica graph impact retrieval efficiency or memory footprint?** No. The replica graph is a **logical concept** used to disentangle semantic roles during memory summarization. It does not result in physically duplicating memory content or embedding storage.
>
> **A Qualitative Example.** To fully address your concern, we further present a qualitative example to demonstrate the flexibility enabled by Ego-Centric Disentanglement. The example is drawn from the long-form novel _Jane Eyre_ and involves a comprehensive question that requires integrating multiple semantically distant chunks of information, as detailed below.
> ```
> Question: Why does Jane leave Thornfield after her wedding is interrupted?
> Evidence Pieces:
> - Jane discovers that Rochester is already married (factual revelation).
> - She experiences intense internal moral conflict about staying (psychological theme).
> - She reflects on religious and spiritual values (spiritual/ethical dimension).
> - She asserts her personal independence and dignity (character development arc).
>
> [Without Flexibility]
> Response: Jane leaves Thornfield because she finds out Rochester is already married. ❌
> (This answer is semantically shallow, missing Jane’s inner struggle, moral reasoning, and identity preservation---these themes are essential to the novel’s integrity.)
>
> [With Flexibility]
> Response: Jane leaves Thornfield not only because of Rochester’s hidden marriage, but also because staying would betray her moral and spiritual convictions. Despite her love, she chooses self-respect and integrity, fearing the loss of her identity. ✔️
> ```
> >### **Q6: Dynamic robustness of CAM to batch order.**
>
> Following your advice, we conduct a new experimental analysis to evaluate CAM's dynamic robustness to **_random_** batch order. We report the **memory statistics** (branching factor) / **inference performance** (F1 for MH-RAG, R-L for others) under each configuration, which are **averaged over five runs**.
>
> |Method|NovelQA|QMSum|MH-RAG|ODSum-Story|
> |:-:|:-:|:-:|:-:|:-:|
> |MemTree-1|3.5±1.8 / 18.6±3.2|4.2±1.6 / 20.2±2.7|3.2±1.9 / 69.7±2.4|4.6±2.2 / 20.4±3.5|
> |CAM-1|5.4±0.9 / 23.3±1.5|5.7±0.8 / 24.2±1.0|6.5±0.8 / 74.3±1.2|6.3±1.2 / 22.8±1.4|
> |CAM-100|5.9±0.5 / 24.2±1.0|6.2±0.8 / 25.3±0.7|7.8±0.5 / 75.0±1.1|7.3±0.9 / 23.5±0.8|
> |CAM-300|5.8±0.5 / 24.7±0.8|6.4±0.5 / 25.7±0.5|7.5±0.6 / 76.2±0.8|7.0±0.7 / 24.4±0.7|
>
> We observe that:
> (1) Even with batch size = 1, CAM yields more stable memory structures and inference performance than MemTree, confirming the effectiveness of flexible assimilation and dynamic accommodation in our framework.
> (2) As the batch size increases, CAM inherently benefits from a broader contextual view, thereby further reducing the memory variations across different batch orders.
> >### **Q7: Hyperparameter sensitivity analysis.**
>
> To assess the robustness of CAM with respect to the key hyperparameters (i.e., $\alpha$, $\sigma$, and $\theta$), we conduct a sensitivity analysis using 100 randomly sampled validation instances from each dataset. The performance results (**averaged over three runs**) across different hyperparameter combinations are summarized below:
>
> |$\alpha$|$\sigma$|$\theta$|NovelQA (R-L)|QMSum (R-L)|FABLES (F1)|MH-RAG (F1)|ODSum-Story (R-L)|ODSum-Meeting (R-L)|Avg.|
> |:-:|:-:|:-:|:-:|:-:|:-:|:-:|:-:|:-:|:-:|
> |0.5|1.0|0.5|25.5|24.7|**53.2**|74.2|24.5|21.3|37.2|
> |0.5|1.0|0.7|**26.1**|25.3|52.9|74.7|25.3|22.6|37.8|
> |0.5|2.0|0.5|24.3|24.2|52.6|75.3|24.4|21.5|37.1|
> |0.5|2.0|0.7|24.8|24.3|52.3|75.5|24.8|22.9|37.4|
> |0.7|1.0|0.5|25.2|26.0|52.7|77.1|24.9|21.8|38.0|
> |0.7|1.0|0.7|25.4|26.5|52.5|77.5|**25.5**|23.2|**38.4**|
> |0.7|2.0|0.5|24.0|25.8|52.2|76.4|24.6|22.4|37.6|
> |0.7|2.0|0.7|24.4|26.2|51.8|77.0|25.1|**23.5**|38.0|
> |0.9|1.0|0.5|23.1|26.0|51.5|77.6|24.3|21.0|37.3|
> |0.9|1.0|0.7|23.5|**26.7**|51.1|**78.3**|24.7|22.6|37.8|
> |0.9|2.0|0.5|22.9|26.2|51.4|77.9|24.2|20.9|37.2|
> |0.9|2.0|0.7|23.3|26.5|50.8|78.2|24.5|21.7|37.5|
>
> We observe that:
> (1) For narrative datasets (e.g., NovelQA and FABLES), incorporating positional proximity (lower $\alpha$) is beneficial, as the adjacent sentences are semantically cohesive.
> (2) The performance is relatively robust to $\sigma$ across datasets, and the sharper decay setting ($\sigma$ = 1.0) generally outperforms the flatter alternative ($\sigma$ = 2.0).
> (3) $\theta$ controls the density of the foundational semantic network and shows the dataset-specific sensitivity.
>
> **We sincerely promise to incorporate the above experiments and discussions into our final version. We hope that these responses have addressed your concerns, and would greatly appreciate it if you could kindly raise the scores accordingly. Your support really means a great deal to us.**

---

> > ### Comment · Reviewer_vP7d · 2025-08-09
> > **Acknowledgement of Rebuttal and Confirmation of Score**
> >
> > I appreciate the authors’ thorough and well‐structured rebuttal, which directly addresses each of my earlier concerns with additional experiments, sensitivity analyses, and clarifications. In particular, the new retrieval comparisons, batch‐order robustness tests, and hyperparameter sensitivity study strengthen the empirical support for CAM’s claims. While these responses improve my confidence in the technical soundness and completeness of the work, my overall evaluation remains unchanged, as some limitations (e.g., handling contradictory information) remain as future work rather than being resolved in the current submission.

---

> ### Author Response · Authors · 2025-08-07
>
> Dear Reviewer vP7d,
>
> Thank you for taking the time to acknowledge our rebuttal.
>
> During the rebuttal phase, we have made great efforts to address each of your concerns. In particular, we would like to kindly ask whether our responses have addressed your concerns regarding **the additional analysis on inference efficiency and dynamic robustness**.
>
> If you have any further questions or suggestions, we would be happy to continue the discussion. Your support really means a great deal to us, and we sincerely look forward to your feedback.
>
> Best regards,
> Authors

---

### Official Review · Reviewer_4RE6 · 2025-06-25

**Clarity:** 3
**Significance:** 2
**Originality:** 3
**Rating:** 4
**Confidence:** 3

**Summary:**

This paper proposes CAM (Constructivist Agentic Memory) and identify three key traits for effective agentic memory. CAM further developed a prototype and conduct experiments on various reading comprehension tasks. The results demonstrate the improvement of CAM on effectiveness and efficiency.

**Questions:**

1. I noticed in Table 3, there are some ablations conducted, I'm wondering whats the relationship between these ablations and the proposed principles.
2. As to the implementation, I'm wondering whether the author has plans to release / open-source the prototype.

**Ethical Concerns:**

["NO or VERY MINOR ethics concerns only"]

**Final Justification:**

The rebuttal has resolved my concerns properly.

**Limitations:**

yes

**Paper Formatting Concerns:**

No concerns.

**Quality:**

3

**Strengths And Weaknesses:**

Strength:

The paper studied an important and emerging problem and proposed several principles to address the problem. The resulting prototype system works well on a wide range of tasks.

Weakness:

My major concern is that the main claim of this paper focuses on the three proposed principles, some of which is implemented in existing systems. The author only conduct experiments with the implemented prototye using all three principles and no ablation is conducted on these principles. However, to properly support the proposed principles are necessary and important, detailed ablation studies are needed to analyze the benefits and impacts of those principles.

Another weakness is on the additional hyper-parameters. Since the proposed method introduces multiple hyper-parameters, it is unclear how sensitive the model is to the additional hyper-parameters, how much of the performance improvements attributes to their tuning.

----
i've read the rebuttal and updated my scores.

---

> ### Author Rebuttal · Authors · 2025-07-30
>
> Thanks for your careful and insightful comments. Below, we provide point-by-point responses to each concern raised.
>
> >### **Q1: Clarification on the ablations of the proposed principles.**
>
> We would like to respectfully clarify that we have conducted comprehensive ablation studies to evaluate the contribution of the three proposed principles (i.e., Hierarchy, Flexibility, and Dynamicity), as detailed in **Sections 5.3 and 5.4**:
> - For **Hierarchy** and **Flexibility**, we present two ablation variants (**_w/o Hierarchy_** and **_w/o Flexibility_**) in **Table 3**, evaluated on four long-context reading comprehension benchmarks.
> - The **_w/o Hierarchy_** variant omits the hierarchical clustering stage entirely and relies solely on the foundational semantic network.
> - The **_w/o Flexibility_** variant bypasses ego-centric disentanglement and directly applies non-overlapping label propagation for clustering.
> - For your convenience, we **re-include their comparison with the full CAM below**. We can see that both variants result in clear performance degradation, thereby confirming the importance of hierarchy and flexibility in our design principles.
>
> |Methods|NovelQA (R-L / ACC-L)|QMSum (R-L / ACC-L)|MH-RAG (EM / F1)|ODSum-Story (R-L / ACC-L)|
> |:----------:|:----------------:|:----------------------:|:-------------------------:|:-------------------------:|
> |**Full CAM**|**25.4 / 52.3**|**26.5 / 57.6**|**72.8 / 77.5**|**25.5 / 54.6**|
> |_w/o Hierarchy_|22.9 / 46.7|23.2 / 46.6|68.5 / 73.4|23.5 / 49.3|
> |_w/o Flexibility_|24.2 / 50.3|24.7 / 51.5|70.3 / 75.1|24.7 / 51.2|
>
> - Moreover, **the benefits of Dynamicity are extensively investigated in Section 5.3**.
> - Specifically, **Figure 3** illustrates how the dynamicity of CAM contributes to both **processing efficiency** and **inference stability**.
> - In terms of **processing efficiency** (Figure 3a), CAM can dynamically integrate newly arrived chunks in batches, offering **over 4× speedup** compared to methods constrained to offline (RAPTOR, GraphRAG) or entry-wise online processing (MemTree).
> - In terms of **inference stability** (Figure 3b), CAM maintains **stable and competitive performance** across varying batch sizes. These results collectively validate the benefits and necessity of our dynamicity principle.
>
>
>
> >### **Q2: Hyperparameter sensitivity analysis.**
>
> Thank you for raising this important point. To assess the robustness of CAM with respect to the key hyperparameters (i.e., $\alpha$, $\sigma$, and $\theta$), we conduct a sensitivity analysis using 100 randomly sampled validation instances from each dataset. We perform grid search over the following values: $\alpha\in$ {$0.5,0.7,0.9$}, $\sigma\in$ {$1.0,2.0$}, and $\theta\in$ {$0.5,0.7$}. The performance results (**averaged over three runs**) across different hyperparameter combinations are summarized below:
>
> |$\alpha$|$\sigma$|$\theta$|NovelQA (R-L)|QMSum (R-L)|FABLES (F1)|MH-RAG (F1)|ODSum-Story (R-L)|ODSum-Meeting (R-L)|Avg.|
> |:----------:|:----------------:|:----------------------:|:-------------------------:|:-------------------------:|:-------------------------:|:-------------------------:|:-------------------------:|:-------------------------:|:-------------------------:|
> |0.5|1.0|0.5|25.5|24.7|**53.2**|74.2|24.5|21.3|37.2|
> |0.5|1.0|0.7|**26.1**|25.3|52.9|74.7|25.3|22.6|37.8|
> |0.5|2.0|0.5|24.3|24.2|52.6|75.3|24.4|21.5|37.1|
> |0.5|2.0|0.7|24.8|24.3|52.3|75.5|24.8|22.9|37.4|
> |0.7|1.0|0.5|25.2|26.0|52.7|77.1|24.9|21.8|38.0|
> |0.7|1.0|0.7|25.4|26.5|52.5|77.5|**25.5**|23.2|**38.4**|
> |0.7|2.0|0.5|24.0|25.8|52.2|76.4|24.6|22.4|37.6|
> |0.7|2.0|0.7|24.4|26.2|51.8|77.0|25.1|**23.5**|38.0|
> |0.9|1.0|0.5|23.1|26.0|51.5|77.6|24.3|21.0|37.3|
> |0.9|1.0|0.7|23.5|**26.7**|51.1|**78.3**|24.7|22.6|37.8|
> |0.9|2.0|0.5|22.9|26.2|51.4|77.9|24.2|20.9|37.2|
> |0.9|2.0|0.7|23.3|26.5|50.8|78.2|24.5|21.7|37.5|
>
> We observe that:
> (1) For narrative datasets (e.g., NovelQA and FABLES), incorporating positional proximity (lower $\alpha$) is beneficial, as the adjacent sentences are semantically cohesive.
> (2) The performance is relatively robust to $\sigma$ across datasets, and the sharper decay setting ($\sigma$ = 1.0) generally outperforms the flatter alternative ($\sigma$ = 2.0).
> (3) $\theta$ controls the density of the foundational semantic network and shows the dataset-specific sensitivity.
>
> >### **Q3: The relationship between the ablations in Table 3 and the proposed principles.**
>
> As you noticed, Table 3 comprehensively presents the performance of CAM with **different configurations** and **two ablation variants**.
>
> As described in Section 5.4, we group the rows into four categories for clarity:
> - **LLM Backbones:** We evaluate CAM using different language models, including the default commercial model **GPT-4o-mini** and two popular open-source alternatives: **Llama-3.1-8B-Instruct** and **Qwen2.5-7B-Instruct**. The results suggest that CAM's core strengths primarily stem from the effective memory design, rather than relying on expensive LLM backbones.
> - **Embedding Models:** We assess the effect of different embedding encoders, including **text-embedding-3-small** (default), **text-embedding-3-large**, and **E5-Mistral-7B-Instruct**. The results show that the use of more advanced embedding models can improve the performance of CAM, since they deliver more accurate similarity measurements for the construction of memory foundational networks.
> - **Retrieval Strategy:** This section compares our **Prune-and-Grow** strategy with two alternatives: **Hierarchical Traversal** and **Global Retrieval**. We observe that both alternatives lead to lower performance, demonstrating the effectiveness of our Prune-and-Grow strategy.
> - **Ablation Variants:** As elaborated in our response to your Q1, these two variants are designed to demonstrate the importance of the Hierarchy and Flexibility principles in CAM's design. The **_w/o Hierarchy_** variant removes the hierarchical abstraction, and the **_w/o Flexibility_** variant disables the ego-centric disentanglement. Both variants result in consistent performance degradation, thereby supporting the benefits and rationality of these two principle designs.
>
> >### **Q4: Plan to release the source code of the prototype.**
>
> Sincerely thanks for your interest. We have provided the implementation code in an anonymous repository (linked in the footnote on the first page) to support reproducibility during the review process. Upon publication, we will open-source the full codebase, and further develop it into a general-purpose constructivist memory framework to benefit the broader research community.
>
> **We sincerely promise to incorporate the above experiments and discussions into our final version. We hope that these responses have addressed your concerns, and would greatly appreciate it if you could kindly raise the scores accordingly. Your support really means a great deal to us.**

---

> ### Author Response · Authors · 2025-08-05
>
> Dear Reviewer 4RE6,
>
> Thank you very much for taking the time to acknowledge our rebuttal.
>
> During the rebuttal phase, we have made great efforts to address each of your concerns. In particular, we would like to kindly ask **whether our responses have clarified your misunderstandings regarding our ablation studies**.
>
> If you have any further questions or suggestions, we would be happy to continue the discussion. Your support really means a great deal to us, and we sincerely look forward to your feedback.
>
> Best regards,
> Authors

---

> > ### Comment · Reviewer_4RE6 · 2025-08-06
> >
> > As updated in the review, I've read the rebuttal and increased the score to 4.

---

> ### Author Response · Authors · 2025-08-06
>
> Dear Reviewer 4RE6,
>
> Thank you very much for your feedback and support on our paper. We sincerely promise to include the above description and analysis in our final version.
>
> Best regards,
> Authors

---

### Official Review · Reviewer_mkPe · 2025-07-02

**Clarity:** 3
**Significance:** 2
**Originality:** 2
**Rating:** 4
**Confidence:** 4

**Summary:**

This paper presents Constructivist Agentic Memory (CAM), a memory system for LLMs inspired by Piaget’s theory. It uses structured knowledge units (schemata), adapts flexibly to new input (assimilation) and updates when needed (accommodation). CAM builds memory using overlapping clusters and a "prune-and-grow" retrieval method which helps it understand long texts more effectively and efficiently.

**Questions:**

Please see the weaknesses.

**Ethical Concerns:**

["NO or VERY MINOR ethics concerns only"]

**Final Justification:**

The authors have conducted many new experiments during the rebuttal phase, and most of the experimental results they presented are new. Incorporating these additional results into the paper would require substantial writing revisions, as they would need to be fully integrated into the relevant sections with updated analyses and discussions.

**Limitations:**

yes

**Quality:**

3

**Strengths And Weaknesses:**

**Strengths:**

The paper presents a well-structured and theory-based design for LLM memory, inspired by Jean Piaget’s Constructivist Theory, which serves as a guide for building agent-like memory. CAM not only reaches top performance on various long-text reading tasks but also shows strong efficiency, particularly when handling data in batches during online processing.


**Weaknesses:**

1. The Foundational Network Expansion and Ego-Centric Disentanglement steps rely on several hyperparameters (α, σ, top-k, θ), which greatly affect graph quality and performance. Poor tuning can lead to weak graph structures, ineffective ego-networks and scalability issues - especially in large or diverse texts. While these parameters are common, the paper lacks analysis on their sensitivity and tuning. The paper would benefit from a more thorough analysis of the sensitivity of CAM's performance to these hyperparameters. It would be helpful to explore automatic adjustment methods such as reinforcement or meta-learning to reduce manual tuning and improve adaptability.

2. The Dynamic Accommodation process (Sections 3.3 and 4.1.3) uses "local-first overlapping clustering" to efficiently update memory without reworking everything. While this is efficient, it may risk catastrophic forgetting or information loss if the "localized adjustments" are insufficient or misdirected. If new input changes the overall context or makes earlier summaries outdated, local updates alone may not fix the memory structure. For example, if a document shifts themes halfway through, the system might fail to adjust higher-level summaries properly. A possible improvement would be to add periodic lightweight global coherence checks or introduce mechanisms to track and reconcile significant contextual shifts to prevent information decay.

3. The process of combining nodes into clear concise summaries using LLM-based text summarization (lines 239-240) is key to building the hierarchical memory. However, the paper does not explain much about how this summarization is done - like what prompts are used, LLM settings, length limits or how quality is ensured. The quality of these summaries directly affects how well the higher-level nodes work and how effective retrieval is. If the summaries are poor, contain errors or miss important points, the whole memory system could suffer. The paper should provide more details on the summarization method including the prompts, any filtering or editing steps and an evaluation of summary quality. It would also help to compare different summarization approaches (like extractive vs. abstractive) or different LLMs to show how these choices impact overall results.

4. The Memory Retrieval phase (Section 4.2) relies heavily on an LLM to pick helpful nodes and expand the activation set using a "Prune-and-Grow" method, which mimics human thinking. However, this makes the process dependent on the LLM’s judgment - if it selects wrong nodes or goes off track, the final answer may be wrong or incomplete. Current metrics might not catch these subtle issues. Also, this iterative approach can be slow and costly for large memories or complex queries, which could limit real-time use. The paper should test how robust this LLM-driven retrieval is, maybe by comparing it to simpler non-LLM-driven graph traversal methods. While ACC-L is a good metric, qualitative analysis of retrieval paths and LLM reasoning tracebacks could provide deeper insights into potential biases or hallucination issues during the associative exploration.

5. The paper compares CAM with other memory systems, but LLM memory design is evolving fast. It mainly focuses on hierarchical clustering and knowledge graphs, while newer memory types like episodic or hybrid systems are emerging. Calling CAM a broad "systematic design principle" may be too strong, since it centers on a specific structure. To support this claim, the paper could briefly explore how Constructivist Theory might also guide other memory types. Acknowledging different memory designs and where CAM fits would make the discussion more complete. It would also help to consider how CAM might work with or adapt to these alternative systems.

---

> ### Author Rebuttal · Authors · 2025-07-30
>
> Thanks for your careful and insightful comments. Below, we provide point-by-point responses to each concern raised.
> >### **Q1: Hyperparameter sensitivity analysis.**
>
> To assess the robustness of CAM w.r.t. the key hyperparameters (i.e., $\alpha$, $\sigma$, and $\theta$), we conduct a sensitivity analysis using 100 randomly sampled validation instances from each dataset. Since **top-k** serves as a soft cap to avoid memory over-density and only takes effect under extreme redundancy, we fix it to 10 throughout. For the other three hyperparameters, the results of different combinations (**averaged over three runs**) are reported below:
>
> |$\alpha$|$\sigma$|$\theta$|NovelQA (R-L)|QMSum (R-L)|FABLES (F1)|MH-RAG (F1)|ODSum-Story (R-L)|ODSum-Meeting (R-L)|Avg.|
> |:-:|:-:|:-:|:-:|:-:|:-:|:-:|:-:|:-:|:-:|
> |0.5|1.0|0.5|25.5|24.7|**53.2**|74.2|24.5|21.3|37.2|
> |0.5|1.0|0.7|**26.1**|25.3|52.9|74.7|25.3|22.6|37.8|
> |0.5|2.0|0.5|24.3|24.2|52.6|75.3|24.4|21.5|37.1|
> |0.5|2.0|0.7|24.8|24.3|52.3|75.5|24.8|22.9|37.4|
> |0.7|1.0|0.5|25.2|26.0|52.7|77.1|24.9|21.8|38.0|
> |0.7|1.0|0.7|25.4|26.5|52.5|77.5|**25.5**|23.2|**38.4**|
> |0.7|2.0|0.5|24.0|25.8|52.2|76.4|24.6|22.4|37.6|
> |0.7|2.0|0.7|24.4|26.2|51.8|77.0|25.1|**23.5**|38.0|
> |0.9|1.0|0.5|23.1|26.0|51.5|77.6|24.3|21.0|37.3|
> |0.9|1.0|0.7|23.5|**26.7**|51.1|**78.3**|24.7|22.6|37.8|
> |0.9|2.0|0.5|22.9|26.2|51.4|77.9|24.2|20.9|37.2|
> |0.9|2.0|0.7|23.3|26.5|50.8|78.2|24.5|21.7|37.5|
>
> We observe that:
> (1) For narrative datasets (NovelQA and FABLES), incorporating positional proximity (lower $\alpha$) is beneficial, as the adjacent sentences are semantically cohesive.
> (2) The performance is relatively robust to $\sigma$ across datasets, and the sharper decay setting ($\sigma$ = 1.0) generally outperforms the flatter one ($\sigma$ = 2.0).
> (3) $\theta$ controls the density of the foundational semantic network and shows the dataset-specific sensitivity.
> >### **Q2: Global coherence checking for inconsistent information.**
>
> We agree that the ability to detect and reconcile inconsistent or contradictory information (e.g., when a document transitions to a new theme) is a critical aspect of human cognition. Consistent with prior LLM-based memory systems such as GraphRAG, RAPTOR, and MemTree, our work assumes that source information is generally consistent and focuses on the memory development process. We believe that memory reconciliation is a highly promising direction for future memory research, and will include a clear discussion of this point in our final version.
> >### **Q3: Discussion on LLM-based text summarization.**
>
> **Experimental Setup.** We respectfully clarify that the detailed settings are provided in Section 5.1 and Appendix F: we use **GPT-4o-mini** with **temperature = 0**, and **text-embedding-3-small** as the embedding model for CAM and all baseline methods. Additionally, all LLM prompt templates have been included in the anonymous repository (linked in the footnote on the first page) **since the initial submission**. For the convenience of readers, we will follow your suggestion to add these prompts to the final version.
>
> **LLM-Based Summarization.** Using LLM to generate summaries has become a common practice in current agentic memory systems (ReadAgent, RAPTOR, GraphRAG, and MemTree). As we discussed in the Limitation section (**Appendix I**), we acknowledge that LLM-generated summaries may carry risks of hallucination propagation. This limitation could be mitigated by introducing more agentic behaviors (such as self-questioning and reflection) to reduce the hallucination issue and improve the summary fidelity.
>
> **Comparing Summarization Strategies Across Different LLMs.** We evaluate CAM and RAPTOR under abstractive and extractive summarization strategies. The results (**averaged over three runs**) are reported below:
>
> |Method|LLM|Strategy|NovelQA|QMSum|MH-RAG|ODSum-Story|
> |:-:|:-:|:-:|:-:|:-:|:-:|:-:|
> |RAPTOR|GPT-4o-mini|Ext.|23.7|24.5|73.6|24.0|
> |||Abs.|22.5|25.1|71.3|24.3|
> |CAM|GPT-4o-mini|Ext.|25.4|26.5|77.5|25.5|
> |||Abs.|24.8|26.9|76.2|25.3|
> |RAPTOR|Llama-3.1-8B-Instruct|Ext.|20.6|21.5|69.8|22.5|
> |||Abs.|19.3|21.2|67.3|22.3|
> |CAM|Llama-3.1-8B-Instruct|Ext.|23.7|25.3|74.2|24.3|
> |||Abs.|22.9|25.2|71.6|24.7|
>
> We observe that extractive memory systems generally outperform abstractive ones, as extractive summarization offer greater factual fidelity and lower risk of hallucination, which is important for ensuring accurate retrieval. Moreover, CAM consistently outperforms RAPTOR under both strategies, confirming the effectiveness of our design principles.
> >### **Q4: Concern about LLM robustness and efficiency in memory retrieval.**
>
> **LLM Retrieval Robustness.** While LLMs carries the potential risk of hallucination, they inherently offer strong advantages in semantic association and reasoning, which are valuable in complex long-context scenarios. Many recent IR works [1] also show that LLM-based re-ranking or selection can enhance retrieval accuracy beyond what simple embedding similarity or rule-based methods can offer.
>
> **LLM Retrieval Efficiency.** To further address this concern, we compare Prune-and-Grow retrieval with two graph traversal strategies: **Breadth-First Search** (BFS) and **Personalized PageRank** (PPR). We report the inference performance / average time (seconds) as follows.
>
> |Retrieval Methods|NovelQA|QMSum|MH-RAG|ODSum-Story|
> |:-:|:-:|:-:|:-:|:-:|
> |Prune-and-Grow|**25.4** / 7|**26.5** / 5|**77.5** / 8|**25.5** / 7|
> |BFS|23.3 / 3|24.7 / 2|75.4 / 3|23.5 / 3|
> |PPR|24.0 / 4|25.2 / 3|76.2 / 5|23.9 / 4|
>
> We observe that while the LLM pruning incurs moderate overhead in response time, it brings consistently favorable performance gains beyond the non-LLM variants. Moreover, we respectfully clarify that **CAM is inherently compatible with various retrieval methods**, allowing users to choose the most suitable one based on their application needs.
> >### **Q5: Qualitative analysis.**
>
> **A Qualitative Example.** We follow your suggestion to analyze a representative question involving the long-form novel _Jane Eyre_ (from the NovelQA dataset), which requires integrating two evidence chunks located over 11K tokens apart.
>
> ```
> Question: It's someone's part to receive the company in Chapter XVII, then what did he/she did after being summoned?
>
> Evidence Chunks:
> - [211] Mrs. Fairfax assumed ... it was her part to receive the company ...
> - [233] Mrs. Fairfax was summoned to give information respecting the resources of the house in shawls, dresses, draperies of any kind ...
>
> Global Retrieval:
> - [159] I should have liked something clearer; but Mrs. Fairfax either could not ... ❌
> - [209] The party were expected to arrive on ... ❌
> - [211] Mrs. Fairfax assumed ... it was her part to receive the company ... ✔️
> - [653] Jane observes Mr. Rochester's strange behavior and Mrs. Fairfax ... (Summary node covering [159], [211]) ❌
> - [701] Leah and Mrs. Fairfax are preparing for company at Thornfield. There is a sense of ... (Summary node covering [209], [211]) ✔️
>
> Direct LLM Reasoning (without Prune-and-Grow):
> She took the company on a tour of the gardens and the surrounding estate. ❌
> (This hallucinated answer is plausible but incorrect, as Chunk [233], which contains the actual answer, is missing)
>
> CAM’s Prune-and-Grow Trace:
> [Pruning Step 1] Chunks [211] and [701] are retained based on LLM judgment.
> [Growing Step 1] Neighbors are explored to form the candidate set that includes [233].
> [Pruning Step 2] The LLM selects [233] and retains [211] as contextually important.
> [Growing Step 2] Neighbors are explored to form the candidate set.
> [Pruning Step 3] The activation set stabilizes with [211] and [233].
>
> Final LLM Reasoning (with Prune-and-Grow):
> She gave information respecting the resources of the house in shawls, dresses, draperies of any kind. ✔️
> (This answer is fully grounded and faithful to the original text)
> ```
>
> **Error Analysis.** To fully address your concern，we further provide a detailed error analysis of 100 errors made by CAM on the NovelQA dataset. These errors can be categorized into three main types:
> - **_LLM Reasoning Errors_**: All relevant nodes were retrieved, but the LLM failed to compose or interpret them correctly, typically due to distraction from less relevant context or hallucinated inference.
> - **_LLM Selection Errors_**: Relevant nodes are present in the candidate pool but not selected into the final activation set.
> - **_Retrieval Coverage Failures_**: Relevant nodes are not explored at all. This often occurs when the input question is ambiguous or underspecified, making initial localization difficult despite the presence of relevant evidence.
>
> |Error Type|Error Percentage (%)|
> |:-:|:-:|
> |LLM Reasoning Errors|45%|
> |LLM Selection Errors|26%|
> |Retrieval Coverage Failures|29%|
>
> We will include these in our final version to provide deeper insight into CAM’s failure modes.
> >### **Q6: Positioning constructivist theory within the broader landscape?**
>
> We appreciate the insightful comment. While CAM centers around a hierarchical structure, the constructivist perspective is not inherently limited to this form. At its core, constructivist theory highlights two crucial memory operations (assimilation and accommodation) over (hybrid-)structured schemata, where the structure may vary in type. This idea can naturally inform the design of episodic or hybrid memory systems. For example, episodic memory systems may benefit from flexible assimilation by associating a single experience with multiple semantic cues, while hybrid systems may use accommodation mechanisms to reconcile conflicting memory streams. We promise to include more discussions in our final version to better position the constructivist design principles within the broader landscape of memory types.
>
> **We sincerely hope that our responses have addressed your concerns. Your support really means a great deal to us.**
>
> [1] Zhu et al. _Large language models for information retrieval: A survey_. 2024.

---

> > ### Comment · Reviewer_mkPe · 2025-08-04
> > **Response to rebuttal**
> >
> > Thank you for your response. I think the paper needs significant revisions. I have decided to keep my current scores.

---

> ### Author Response · Authors · 2025-08-04
>
> Dear Reviewer mkPe,
>
> Thank you very much for your feedback and acknowledgement.
>
> During the rebuttal phase, we have made great efforts to address each of your concerns. **We sincerely promise to incorporate the above discussions into our final version**, including analysis on hyperparameter sensitivity, retrieval efficiency, and qualitative results. In addition, we will also expand the Limitation section to discuss the potential drawbacks of using LLMs for memory summarization, which is the de facto standard in current memory research, and outline directions for future works.
>
> If you have any further questions or suggestions, we would be happy to continue the discussion. We sincerely thank you for your support.
>
> Best regards,
> Authors

---

### Official Review · Reviewer_FDCi · 2025-07-03

**Clarity:** 2
**Significance:** 3
**Originality:** 3
**Rating:** 5
**Confidence:** 4

**Summary:**

This work introduces a LLM-based agentic approach for long context comprehension, with a focus on the systematic design of the agentic memory/index. Inspired by cognitive theories, the proposed memory mechanism features the bottom-up structured schemata, the flexible integration of new information without drastic structural changes, and dynamic redistribution of units and structures during integration. Overall, the structured memory consists of original text chunks, as well as consolidated multi-level abstractions/summaries. During inference, relevant nodes are first retrieved by embedding similarity, and then further selected by LLMs based on the node structures and semantics, serving as the context for question answering.

Experiments are performed on single- and multi-document comprehension tasks, including QA, summarization, and claim verification. Results suggest that the proposed method CAM could achieve better performance and efficiency across all tasks.

**Questions:**

See weaknesses.

**Ethical Concerns:**

["NO or VERY MINOR ethics concerns only"]

**Final Justification:**

I have increased my score from 4 to 5, as my initial concerns have been addressed:

1. The authors have made efforts incorporating more long context comprehension datasets for evaluation, and the proposed method is shown good performance against several baselines.

2. More insightful analysis and qualitative examples are provided on how the proposed method tackles complex queries, which deepens the understanding and clarity of this work.

**Limitations:**

yes

**Quality:**

3

**Strengths And Weaknesses:**

##### Strengths

- A systemetic design of agentic memory is proposed with effective implementation, which supports both offline and online indexing.
- The proposed approach is shown improvement on different long-context tasks.
- Useful features in building memory system are identified, with experiment and ablation supports (e.g., structured memory, many-to-many mapping, memory retrieval approach etc.)

##### Weakness

My main concerns are regarding evaluation: as reading comprehension is a core aspect to evaluate for this work, the selection of datasets is rather unconventional. There are many datasets for long-context or multi-document comprehension that are commonly seen in related papers. For example, one of the baseline adopted in Sec 5.1 is RAPTOR, and the paper of RAPTOR evaluates on NarrativeQA, QASPER, and QuALITY. However, this work adopts none of these datasets. Evaluation on more comprehension datasets is strongly suggested to strengthen the significance and scope of this work.
  - For long context QA dataset: NarrativeQA, LooGLE, ∞ Bench, DetectiveQA, etc.
  - For long context fact verification: NoCha
  - For multi-document comprehesion: HotpotQA, 2WikiMultiHopQA, LV-Eval, MuSiQue

In addition, there lacks in-depth analysis comparing the proposed approach with prior RAG methods, e.g. what information or what types of queries can only be answered by the proposed methods, a qualitative example on the advantages of the proposed Ego-Centric Disentanglement.

Section 4 is generally hard to follow. It will be better to have an illustrative of a simple example/walkthrough explaining 4.1.2 and 4.1.3.

---

> ### Author Rebuttal · Authors · 2025-07-30
>
> Thanks for your careful and insightful comments. Below, we provide point-by-point responses to each concern raised.
>
> >### **Q1: Evaluation on more reading comprehension datasets.**
>
> **Dataset Selection Rationale.** Our selection of evaluation datasets is guided by three criteria:
> - **_Recency of Release_**: The selected datasets (e.g., NovelQA from ICLR 2025 [1], FABLES from COLM 2024 [2]) are newly released and absent from LLM pretraining corpora, thereby minimizing the risk of test-time data leakage and ensuring a clean evaluation environment. In contrast, many conventional benchmarks (e.g., NarrativeQA, QASPER, QuALITY) have been released over three years, and thus pose a higher risk of data contamination [1-2].
> - **_Long-Context Property_**: These datasets feature extremely long input texts, often exceeding 150K tokens, which are ideal for evaluating the scalability of agentic memory systems. By comparison, conventional benchmarks such as QuALITY and NarrativeQA feature relatively short contexts, with average lengths around 5K and 50K tokens respectively [1-2].
> - **_Task Diversity_**: We deliberately include a broad range of task types—question answering, query-based summarization, and claim verification—to comprehensively test the generalizability of CAM.
>
> **Evaluation on MHQA Datasets (Appendix G).** To align with standard practices, we have already included evaluations on three widely-used multi-hop QA benchmarks---**HotpotQA**, **2WikiMultiHopQA**, and **MuSiQue**---as presented in **Appendix G**. CAM achieves superior performance across the three datasets, confirming that our design not only excels on long narrative inputs but also generalizes well to classical MHQA settings.
>
> **Evaluation on More Datasets.** To further strengthen the scope of our work, we follow your suggestion to conduct more experiments on **six conventional benchmarks**. We compare CAM against four representative baselines using consistent experimental setups (same LLMs, embedding models, etc.) to ensure fairness. For your convenience, we also **re-include results on the MHQA datasets** below. These comprehensive results confirm CAM's robust performance across tasks and domains, significantly broadening the scope of our work.
>
> |Methods|NarrativeQA (R-L)|LooGLE (R-L)|∞-Bench (R-L)|DetectiveQA (R-L)|NoCha (Pair Acc)|LV-Eval (F1)|HotpotQA (F1)|2Wiki (F1)|MuSiQue (F1)|Average|
> |:----------:|:----------------:|:----------------------:|:-------------------------:|:-------------------------:|:-------------------------:|:-------------------------:|:-------------------------:| :-------------------------:|:-------------------------:|:-------------------------:|
> |ReadAgent|22.9|35.4|23.2|28.2|38.3|10.8|65.1|59.6|30.9|34.9|
> |RAPTOR|26.6|42.7|29.5|31.4|46.8|14.6|73.4|66.2|34.8|40.7|
> |GraphRAG|28.3|39.5|30.4|31.1|44.0|15.8|72.5|68.7|35.2|40.6|
> |MemTree|25.7|37.2|25.5|29.4|41.7|12.1|70.8|65.0|31.6|37.7|
> |**CAM**|**30.5**|**44.1**|**32.7**|**32.3**|**48.1**|**16.4**|**75.4**|**70.2**|**37.2**|**43.0**|
>
>
>
> >### **Q2: In-depth fine-grained performance analysis.**
>
> We appreciate your suggestion to provide a more nuanced comparison of CAM against prior methods. To this end, we perform **fine-grained performance analysis** on the NovelQA dataset by leveraging its available question-type annotations. Specifically, the queries are categorized along two orthogonal axes:
>
> (1) **_Complexity-Based Categories_**, including _Single-hop_ (42.8%), _Multi-hop_ (35.0%), and _Detail_ (22.2%) questions, which reflect the reasoning depth required to arrive at an answer;
>
> (2) **_Aspect-Based Categories_**, covering _Time_, _Meaning_, _Span_, _Setting_, _Relation_, _Character_, and _Plot_, which reflect the semantic focus of each query.
>
> We compare CAM against four representative baselines (ReadAgent, RAPTOR, GraphRAG, MemTree) under identical settings.
>
> **[Performance on Complexity Categories]**
> |Methods|Single-hop (42.8%)|Multi-hop (35.0%)|Detail (22.2%)|Average|
> |:----------:|:----------------:|:----------------------:|:-------------------------:|:-------------------------:|
> |ReadAgent|27.3|13.2|18.5|20.4|
> |RAPTOR|30.5|17.3|20.7|23.7|
> |GraphRAG|29.4|16.7|18.4|22.5|
> |MemTree|28.7|14.4|17.5|21.2|
> |**CAM**|**31.6**|**20.1**|**21.8**|**25.4**|
>
> **[Performance on Aspect Categories]**
> |Methods|Times (20.1%)|Meaning (15.9%)|Span (1.5%)|Setting (11.5%)|Relation (7.1%)|Character (18.3%)|Plot (25.6%)|Average|
> |:----------:|:----------------:|:----------------------:|:-------------------------:|:-------------------------:|:-------------------------:|:-------------------------:|:-------------------------:|:-------------------------:|
> |ReadAgent|15.6|15.3|14.1|22.5|20.2|23.7|24.4|20.4|
> |RAPTOR|19.8|18.4|15.8|26.3|24.9|26.1|27.3|23.7|
> |GraphRAG|19.1|17.6|15.5|24.7|26.2|23.5|26.8|22.5|
> |MemTree|14.2|16.9|14.7|23.5|24.2|24.8|25.3|21.2|
> |**CAM**|**21.3**|**20.6**|**19.4**|**28.5**|_25.8_|**27.9**|**28.7**|**25.4**|
>
> The results demonstrate that CAM exhibits particularly strong gains on complex questions (e.g., Multi-hop, Times, and Span), which require integrating semantically distant evidence and maintaining narrative coherence.
>
>
>
> >### **Q3: A qualitative example on the advantages of Ego-Centric Disentanglement.**
>
> Ego-Centric Disentanglement is central to CAM’s ability to form coherent and nuanced memory abstractions. Its core motivation lies in **capturing the semantic multifacetedness of complex text chunks**---each chunk often relates to multiple themes, topics, or concepts. By replicating nodes based on their local semantic neighborhoods, CAM allows a single chunk to contribute to multiple higher-level abstractions, enabling more comprehensive and context-sensitive summarization.
>
> To fully address your concern, we follow your suggestion and present a qualitative example to demonstrate the flexibility enabled by Ego-Centric Disentanglement. The example is drawn from the long-form novel _Jane Eyre_ and involves a comprehensive question that requires integrating multiple semantically distant chunks of information, as detailed below.
>
> ```
> Question: Why does Jane leave Thornfield after her wedding is interrupted?
> Evidence Pieces:
> - Jane discovers that Rochester is already married (factual revelation).
> - She experiences intense internal moral conflict about staying (psychological theme).
> - She reflects on religious and spiritual values (spiritual/ethical dimension).
> - She asserts her personal independence and dignity (character development arc).
>
> [Without Flexibility]
> Response: Jane leaves Thornfield because she finds out Rochester is already married. ❌
> (This answer is semantically shallow, missing Jane’s inner struggle, moral reasoning, and identity preservation---these themes are essential to the novel’s integrity.)
>
> [With Flexibility]
> Response: Jane leaves Thornfield not only because of Rochester’s hidden marriage, but also because staying would betray her moral and spiritual convictions. Despite her love, she chooses self-respect and integrity, fearing the loss of her identity. ✔️
> ```
>
>
>
> >### **Q4: Improve the clarity of Sections 4.1.2 and 4.1.3 with a simple example.**
>
> Thank you for the suggestion. We will include a clear, step-by-step toy example in the revision to illustrate the processes in Sections 4.1.2 and 4.1.3. This example will be aligned with Figure 2 to better explain how memory nodes are disentangled and updated during the construction process.
>
>
> **We sincerely promise to incorporate the above experiments and discussions into our final version. We hope that these responses have addressed your concerns, and would greatly appreciate it if you could kindly raise the scores accordingly. Your support really means a great deal to us.**
>
> [1] _Novelqa: Benchmarking question answering on documents exceeding 200k tokens_. ICLR 2025.
> [2] _Fables: Evaluating faithfulness and content selection in book-length summarization_. COLM 2024.

---

> ### Author Response · Authors · 2025-08-05
>
> Dear Reviewer FDCi,
>
> Thank you very much for taking the time to acknowledge our rebuttal.
>
> During the rebuttal phase, we have made great efforts to address each of your concerns. In particular, we would like to kindly ask whether our responses have addressed your concerns regarding **the evaluation on more conventional datasets** and **the fine-grained performance analysis**.
>
> If you have any further questions or suggestions, we would be happy to continue the discussion. Your support really means a great deal to us, and we sincerely look forward to your feedback.
>
> Best regards,
> Authors

---

### Note · Authors · 2025-08-11

We sincerely thank all reviewers for their support and feedback. Below, we summarize our main contributions, highlight the issues addressed during the rebuttal, and clarify the scope of our work.
## **Contributions**
- **[Blueprint]** This work is the first to draw inspiration from Jean Piaget's Constructivist Theory, explicitly highlighting three critical properties for memory development: structured schemata, flexible assimilation, and dynamic accommodation.
- **[Prototype]** Based on these principles, we develop a prototype of Constructivist Agentic Memory that significantly enhances the reading comprehension capabilities of LLMs. Its effectiveness is validated across 9 diverse datasets.
## **Rebuttal Highlights**
- **[Broader Evaluation]** In addition to the 9 datasets included in our original submission, we followed Reviewer FDCi's suggestion to evaluate CAM on 6 more conventional datasets, further confirming the generalizability of our design.
- **[Fine-grained Performance]** We provided in-depth fine-grained evaluations, showing that CAM exhibits particularly strong improvements on complex question types.
- **[Hyperparameter Sensitivity]** We reported a detailed sensitivity analysis, showing that CAM is relatively robust to the hyperparameter variations.
- **[Retrieval Strategy]** We clarified that CAM is inherently compatible with various retrieval methods, allowing users to choose the suitable one based on their needs.
- **[Dynamic Robustness]** We tested CAM under random batch orders, showing that CAM exhibits stronger dynamic robustness than the baseline.
- **[Qualitative Analysis]** We also included illustrative examples and error analyses to provide deeper insights into CAM’s strengths and failure cases.
## **Scope Clarification**
Some concerns were raised regarding **the reliability of LLM-based summarization** and **the handling of contradictory information**. We respectfully clarify that:
- **The use of LLMs for memory summarization is aligned with prior works (RAPTOR, GraphRAG, and MemTree)**, and has become a de facto standard in LLM memory research.
- **The handling of contradictory source information is out of the scope of this work**. Our work, following the existing LLM memory systems, operates under the common assumption that source information is generally consistent. Handling conflicting information requires dedicated research efforts, such as new benchmarks and evaluation protocols, which are beyond the scope of this 9-page paper.

---

### Decision · Program_Chairs · 2025-09-17

**Decision:**

Accept (poster)

**Comment:**

This paper proposed a new way to organize the memory of an LLM agent so it can handle very long context effectively and efficiently. The method has a clear root in the Constructivist theory of Jean Piaget. It has three key traits: structured schemata, flexible assimilation and dynamic accommodation. The method works as a hierarchical overlapping summarization and online batch integration. Experiments showed that the method has strong performance in diverse long context tasks. The implementation is also efficient, adding to the paper's practical application value.

Paper strength

Important problem: effective use of long context is an important topic in the application of LLMs.
Solid theoretical root. This is not just a trial-n-error empirical construction but with more structured thinking in the design and backed by a widely understood theory framework.
Strong performance over a variety of tasks. It's not limited to a narrow domain or type of task.
Practical value brought by the efficient implementation.

Paper weakness

Not able to handle contradictory information that feeds into the memory. Although the author has a point that most long context tasks now usually assume the coherence and correctness of the context, it doesn't remove the weakness. Reviewers are not fully convinced by this argument.

Use of LLM to do summarization is a critical dependency on a blackbox-like system. The author is right to say that this is a common practice; the strong performance with existing LLM partly elevates the concern too. AC suggests to get a better understanding on the minimal capability characteristic of a blackbox LLM to support the method, and/or find a way to "certify" a LLM.

Most of the other weakness in the original submission have been addressed by the rebuttal process with factual new information. For example the experiments on more conventional benchmarks, hyperparameter sensitivity, sensitivity to batch order etc. The reviewers are convinced.

Reviewers also called out that the time left for the authors to rewrite the paper with new evidence is quite limited.

Given all information, AC tends to accept the paper, due to the benefit of sharing this principled way to construct agent memory is larger than bearing the unaddressed weakness (given that the weakness will be clearly stated in the final version).

AC appreciates the author's great efforts in answering questions from the reviewers and running additional experiments.